# Impact of seasonal variations in *Plasmodium falciparum* malaria transmission on the surveillance of *pfhrp2* gene deletions

Oliver John Watson[1]*, Robert Verity[1], Azra C Ghani[1], Tini Garske[1], Jane Cunningham[2], Antoinette Tshefu[3], Melchior K Mwandagalirwa[3,4], Steven R Meshnick[4,5], Jonathan B Parr[5], Hannah C Slater[1]

[1]MRC Centre for Global Infectious Disease Analysis, Department of Infectious Disease Epidemiology, Imperial College London, London, United Kingdom; [2]Global Malaria Programme, World Health Organization, Geneva, Switzerland; [3]School of Public Health, University of Kinshasa, Kinshasa, Democratic Republic of the Congo; [4]Department of Epidemiology, Gillings School for Global Public Health, University of North Carolina at Chapel Hill, Chapel Hill, United States; [5]Division of Infectious Diseases, Department of Medicine, School of Medicine, University of North Carolina at Chapel Hill, Chapel Hill, United States

**Abstract** Ten countries have reported *pfhrp2/pfhrp3* gene deletions since the first observation of *pfhrp2*-deleted parasites in 2012. In a previous study (Watson et al., 2017), we characterised the drivers selecting for *pfhrp2/3* deletions and mapped the regions in Africa with the greatest selection pressure. In February 2018, the World Health Organization issued guidance on investigating suspected false-negative rapid diagnostic tests (RDTs) due to *pfhrp2/3* deletions. However, no guidance is provided regarding the timing of investigations. Failure to consider seasonal variation could cause premature decisions to switch to alternative RDTs. In response, we have extended our methods and predict that the prevalence of false-negative RDTs due to *pfhrp2/3* deletions is highest when sampling from younger individuals during the beginning of the rainy season. We conclude by producing a map of the regions impacted by seasonal fluctuations in *pfhrp2/3* deletions and a database identifying optimum sampling intervals to support malaria control programmes.
DOI: https://doi.org/10.7554/eLife.40339.001

*For correspondence:
o.watson15@imperial.ac.uk

Competing interests: The authors declare that no competing interests exist.

## Introduction

Diagnostic testing of suspected malaria cases has more than doubled in the last 15 years, with 75% of suspected cases seeking treatment from the public health sector receiving a diagnostic test in 2017 (*World Health Organization, 2018a*). Much of this progress reflects the increased distribution of rapid diagnostic tests (RDTs), with the most commonly used RDTs targeting the *P. falciparum* protein HRP2 (PfHRP2). In 2014, a review of published reports of *pfhrp2/3* deletions was conducted and included a critical assessment of the comprehensiveness of the diagnostic investigation. (*Cheng et al., 2014*). The findings of this review highlighted a need for a harmonized approach to investigating and confirming or excluding *pfhrp2/3* deletions and called for further studies to determine the prevalence and impact of *pfhrp2/3* gene deletions. Since that review, false-negative RDT results due to *pfhrp2/3* gene deletions have been reported in 10 countries in sub-Saharan Africa (SSA) (*World Health Organization, 2018b*). The frequency of *pfhrp2/3* deletions varies across SSA,

with the highest burden observed in Eritrea where 80.8% of samples from Ghindae Hospital were both *pfhrp2*-negative and *pfhrp3*-negative in 2016 (*Berhane et al., 2018*).

Mathematical modelling has predicted that the continued use of only PfHRP2 RDTs will quickly select for parasites without the *pfhrp2* gene (*Gatton et al., 2017*). This selection pressure occurs due to the misdiagnosis of infections caused by parasites lacking the *pfhrp2* gene, which will subsequently contribute more towards onwards transmission than wild-type parasites that are correctly diagnosed due to the expression of *pfhrp2*. In 2017, we conducted an analysis of the drivers of *pfhrp2* gene deletion selection, identifying the administrative regions in SSA with the greatest potential for selecting for *pfhrp2-deleted* parasites (*Watson et al., 2017*). The regions identified were areas with both a low prevalence of malaria and a high frequency of people seeking treatment and being treated on the basis of PfHRP2-based RDT diagnosis. The precise strength of selection, however, is not known, with other factors such as the rate of non-malarial fevers and non-adherence to RDT outcomes likely to impact the number of misdiagnosed cases receiving treatment.

In February 2018, the World Health Organization (WHO) issued guidance for national malaria control programmes on how to investigate suspected false-negative RDTs with an emphasis on *pfhrp2/3* gene deletions. (*World Health Organization, 2018c*). The primary study outcome to be calculated in the guidance is as follows:

$$\begin{array}{l}\text{Proportion of } \textit{P. falciparum} \text{ cases}\\\text{with false} - \text{negative HRP2 RDT}\\\text{results due to } \textit{pfhrp2/3} \text{ deletions}\end{array} = \dfrac{\begin{array}{c}\text{\# of confirmed } \textit{falciparum} \text{ patients with } \textit{pfhrp2/3}\\\text{gene deletions and HRP2 RDT negative results}\end{array}}{\begin{array}{c}\text{\# of confirmed } \textit{P. falciparum} \text{ cases}\\\text{(by either RDT or microscopy)}\end{array}}$$

The guidance recommends that a national change to non PfHRP2-based RDTs should be made if the estimated proportion of *P. falciparum* cases with false-negative HRP2 RDT results due to *pfhrp2/3* deletions is above 5%. If the estimated proportion is less than 5% the country is recommended to establish a monitoring scheme whereby the study is repeated in two years if the 95% confidence interval does not include 5%, or one year if it does include 5%. The 5% threshold approximates the point at which the number of cases missed due to false-negative PfHRP2-based RDTs caused by *pfhrp2/3* deletions may become greater than the number of cases that would be missed due to the decreased sensitivity of non PfHRP2-based RDTs. The guidance also specifies a sampling scheme to be used when estimating the prevalence of *pfhrp2/3* gene deletions. Samples are to be collected from at least 10 health facilities per province to be tested, with sampling focussed on symptomatic *P. falciparum* patients presenting at the health facilities. All samplings are to be ideally completed within an 8-week period.

The 8-week interval permits for a rapid turnaround and allows for efficient investigations and policy responses. However, the timing of the 8-week interval chosen within a transmission season is important. The chosen interval could lead to estimates of the proportion of *P. falciparum* cases with false-negative HRP2 RDT results due to *pfhrp2/3* deletions that are not representative of the annual average proportion. Subsequently, any recorded estimate may not be predictive of the number of cases that may be misdiagnosed due to *pfhrp2/3* deletions in the years between sampling intervals. For example, an overestimation of the annual average proportion of false-negative RDTs due to *pfhrp2/3* deletions could result in a switch to a less sensitive RDT, resulting in an increase in the number of malaria cases misdiagnosed if the annual average proportion of false-negative RDTs due to *pfhrp2/3* deletions is less than 5%. The alternative RDT may also be both more expensive and complicated to implement. Similarly, an underestimation of the annual average proportion of *P. falciparum* cases with false-negative HRP2 RDT results due to *pfhrp2/3* deletions would result in continued use of an overall less effective test and could provide *pfhrp2/3* deleted parasite populations an opportunity to expand.

In response to these concerns, we extended our original methods (*Watson et al., 2017*) to characterise the impact of seasonal variations in transmission intensity on the proportion of false-negative RDTs due to *pfhrp2*-deleted parasites. We present an extended version of our previous model, which predicts that more false-negative RDTs due to *pfhrp2* gene deletions are observed when monoclonal infections are more prevalent, with the highest proportion observed when sampling

from younger children at the start of the rainy season. We continue to assess how samples collected within an 8-week interval can both over- and underestimate this proportion when compared to the annual average, which reflects the monitoring scheme recommended by the WHO for follow up studies if the outcomes of the original study are inconclusive. Lastly, we map the administrative regions in SSA with the greatest potential for estimates of the proportion of *P. falciparum* cases with false-negative HRP2 RDT results due *to pfhrp2* deletions to be not predictive of the annual average. In addition, we identify the optimum sampling intervals for each level one administrative region, which are most representative of the annual average.

## Results

Using our model, we first explored how the proportion of clinical cases only infected with *pfhrp2*-deleted parasites varies throughout a transmission season. We recorded the proportion of clinical cases that are PfHRP2-negative in four settings (a low and moderate transmission setting with both a low and highly seasonal transmission dynamic), which had a starting *pfhrp2* deletion frequency of 6%. 6% was chosen to reflect our previously estimated frequency of *pfhrp2* deletions prior to the introduction of RDTs in the Democratic Republic of the Congo (DRC) (*Watson et al., 2017*). We initially assumed that the frequency of *pfhrp2* deletions was not increasing over time before considering scenarios in which the selective pressure for *pfhrp2* deletions causes an increase in the population frequency of *phrp2* deletions. This decision allowed for the impact of seasonality on the proportion of clinical cases that are *pfhrp2*-negative to be isolated, before allowing comparisons to scenarios in which the proportion of clinical cases that are *pfhrp2*-negative is increasing also due to changes in the population frequency of *phrp2* deletions.

Our predictions suggest that the misdiagnosis of clinical cases due to *pfhrp2*-negative RDT results is heavily dependent on transmission intensity (*Figure 1*). For the same population frequency of *pfhrp2* gene deletions (*Figure 1Q–T*), the observed proportion of clinical cases that are *pfhrp2*-negative is predicted to be higher in lower transmission settings (*Figure 1I–P*). The annual average proportion of clinical cases that are *pfhrp2*-negative was equal to 5% and 3.25% in the low and moderate transmission setting, respectively. This observation is attributable to the lower rate of superinfection in low transmission settings. The lower rate of superinfection reduces the number of polyclonal infections and increases the chance that an individual is only infected with *pfhrp2*-negative parasites (*Figure 1—figure supplement 1*). When we considered scenarios with a selective advantage for *pfhrp2*-deletions (*Figure 1—figure supplement 2*), the population frequency of *pfhrp2* gene deletions increased over the two years observed (*Figure 1—figure supplement 2Q–T*) with a corresponding increase in the proportion of clinical cases that are *pfhrp2*-negative (*Figure 1—figure supplement 2I–P*).

An increased proportion of individuals only infected with *pfhrp2* gene deletions is predicted to occur at the beginning of the rainy season just before incidence starts to increase. During the rainy season, the observed proportion of cases expected to yield a false-negative RDT due to *pfhrp2*-deleted parasites (PfHRP2-negative) falls, with the lowest proportion observed after the end of the rainy season. These dynamics are more pronounced in highly seasonal transmission regions (*Figure 1B, F, J, N, R, D, H, L, P and T*). In the highly seasonal settings, the observed proportion of clinical cases that are PfHRP2-negative is predicted to fluctuate above and below the 5% threshold for switching RDT provided by the WHO (*Figure 1J, L, N and P*). Smaller fluctuations are seen in less seasonal transmission regions (*Figure 1A, E, I, M, Q, C, G, K, O and S*), with no fluctuations in the observed proportion of clinical cases that are PfHRP2-negative occurring above 5% in the moderate transmission setting (*Figure 1K and O*). Similar patterns were observed in scenarios with an increasing frequency of *pfhrp2*-deletions, with fluctuations in the proportion of clinical cases that were PfHRP2-negative observed in the highly seasonal settings (*Figure 1—figure supplement 2J, L, N and P*). The highest proportion of cases expected to yield a false-negative RDT due to *pfhrp2*-deleted parasites was still observed at the beginning of the rainy season.

The specific 8-week interval during which samples are collected is predicted to impact the observed proportion of false-negative RDTs due to *pfhrp2* gene deletions (*Figure 2*). In a moderate transmission setting, a clear seasonal pattern is predicted (*Figure 2C*), with sampling at the beginning of the transmission seasons resulting in significant overestimation of the annual average proportion of false-negative RDTs. Subsequently, sampling at the end of the rainy season is predicted to

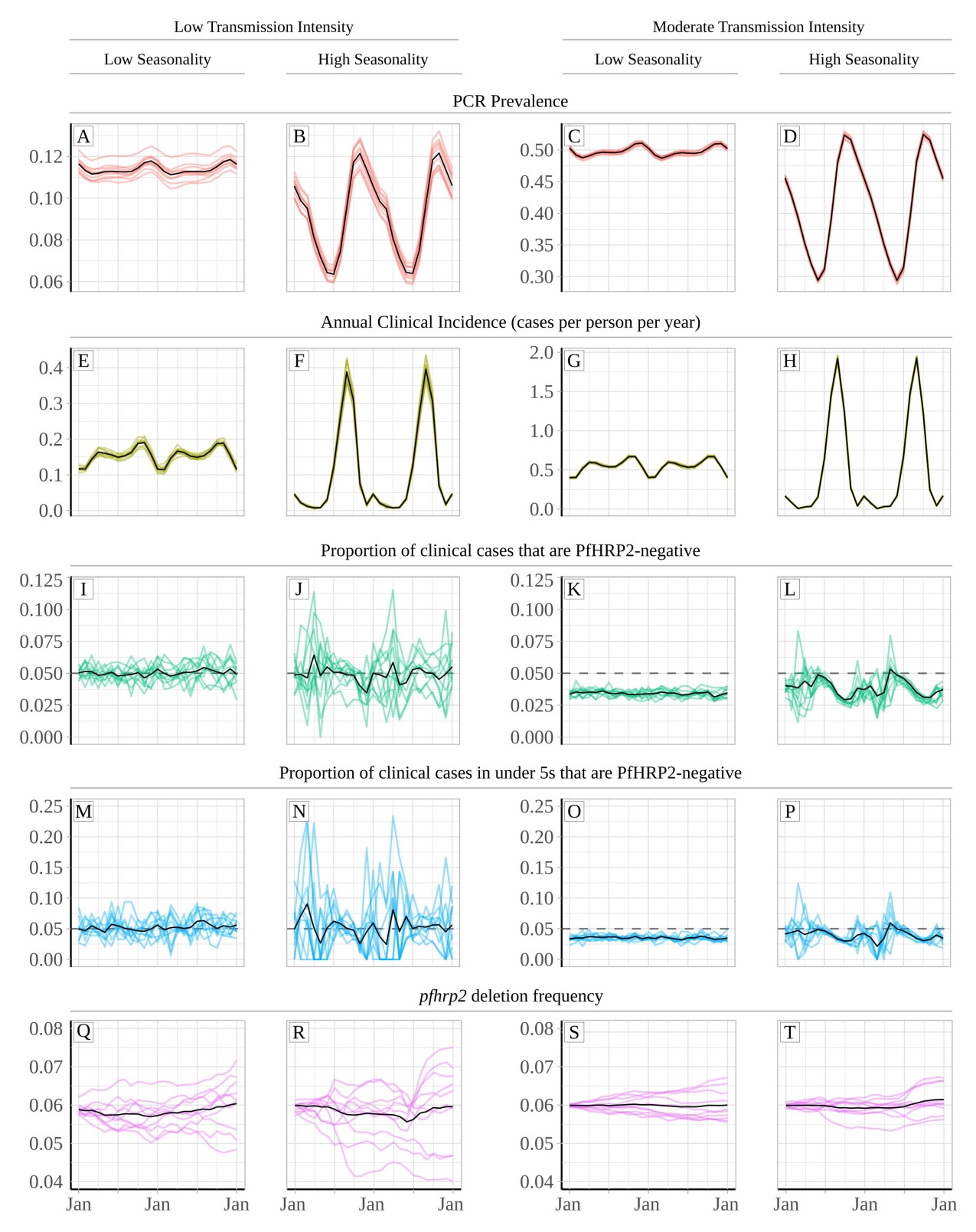

**Figure 1.** Relationship between seasonality, transmission intensity and proportion of clinical cases that are infected with only *pfhrp2*-deleted parasites. Graphs show in (A – D) and (E - H) the model predicted PCR prevalence and annual clinical incidence respectively at both a low and a moderate transmission intensity. In (I – L) and (M - P) the proportion of clinical cases only infected with *pfhrp2*-negative parasites is shown for both the whole population and in children under 5 years old, respectively. Lastly, graphs (Q - T) show the population allele frequency of *pfhrp2* gene deletions, which

*Figure 1 continued on next page*

*Figure 1 continued*

was set equal to 6% at the beginning of each simulation. 10 simulation realisations are shown in each graph, with the mean shown with by the black line. Lastly, the 5% threshold for switching RDT provided by the WHO is shown with the dashed horizontal line in plots (**I – P**).

DOI: https://doi.org/10.7554/eLife.40339.002

The following figure supplements are available for figure 1:

**Figure supplement 1.** Model predicted relationship between clonality of infection in asymptomatic and clinical cases against prevalence of malaria.

DOI: https://doi.org/10.7554/eLife.40339.003

**Figure supplement 2.** Impact of a selective advantage for *pfhrp2*-deleted parasites on the relationship between seasonality, transmission intensity and proportion of clinical cases that are infected with only *pfhrp2*-deleted parasites.

DOI: https://doi.org/10.7554/eLife.40339.004

yield estimates that are most representative of the annual average. In comparison, surveillance in regions with low seasonality is predicted to yield estimates representative of the annual average throughout the transmission season (*Figure 2B and D*). In all settings, using a sampling scheme spanning the entire transmission season produced estimates that accurately estimated the annual average. A moderate increase in the proportion of false-negative RDTs is also predicted when sampling younger individuals, with the same patterns also seen within asymptomatic individuals. This observation reflects the increased probability that children younger than 5 years old yield symptoms after the first infection, due to their comparatively lower acquired clinical immunity. Similar seasonal dynamics were observed in the highly seasonal settings when we considered scenarios with a selective advantage for *pfhrp2*-deletions (*Figure 2—figure supplement 1A and C*).

Using data from a national survey of *pfhrp2* gene deletions in the DRC, we found that the model-predicted outcomes above were similar to those observed in the field (*Figure 3*) (*Parr et al., 2017*). Among 2752 PCR-positive *P. falciparum* cases in the DRC, individuals were more likely to be infected with only *pfhrp2*-negative parasites if the clinical incidence in the month prior to sample collection was lower (p=$4.1 \times 10^{-6}$), and if the individuals were younger (p=0.016). These findings were maintained when comparing across age and transmission groups, with samples collected during periods of lower transmission found to be more likely to be *pfhrp2*-negative in both older and younger age groups (p=$6.6 \times 10^{-5}$ and $5.6 \times 10^{-4}$, respectively). Samples collected in younger individuals were more likely to be *pfhrp2*-negative in both lower and higher transmission groups when compared to older individuals (p=0.06 and 0.06, respectively).

Lastly, we predicted and mapped the potential for estimates collected within 8-week intervals to be unrepresentative of the annual average proportion of false-negative RDTs due to *phrp2* gene deletions across 598 first administrative regions in SSA (*Figure 4*). We predict that 66 regions possess at least one 8-week interval for which a premature switch to a non PfHRP2-based RDT would have been made in more than 75% of simulations (*Figure 4A*) and 29 regions are predicted to possess at least one 8-week interval for which a premature decision to continue using PfHRP2-based RDTs would have been made in more than 75% of simulations (*Figure 4B*). Out of these 29 regions, 25 are also present within the formerly identified 66 regions. The data for each administrative region can be viewed online at the following interactive database https://shiny.dide.imperial.ac.uk/seasonal_hrp2/.

## Discussion

This research characterises the potential for surveillance in highly seasonal areas within sub-Saharan Africa to produce estimates that fail to represent the annual average proportion of *P. falciparum* cases with false-negative HRP2 RDT results due to *pfhrp2* deletions. These findings highlight the impact of both the seasonal timing and the age of individuals sampled when estimating the proportion of false-negative RDTs due to *pfhrp2* deletions. Policy decisions based on the proportion of clinical cases presenting with false-negative RDTs due to *pfhrp2* gene deletions should thus be made with an awareness of the seasonal transmission dynamics of the region considered.

Our modelling predicted that there would be increased observation of false-negative HRP2 RDT results after periods of lower transmission and within younger individuals. This prediction is consistent with a large, nationally representative survey of *pfhrp2*-negative samples among asymptomatic subjects in the DRC (*Parr et al., 2017*). These predictions are also in agreement with other

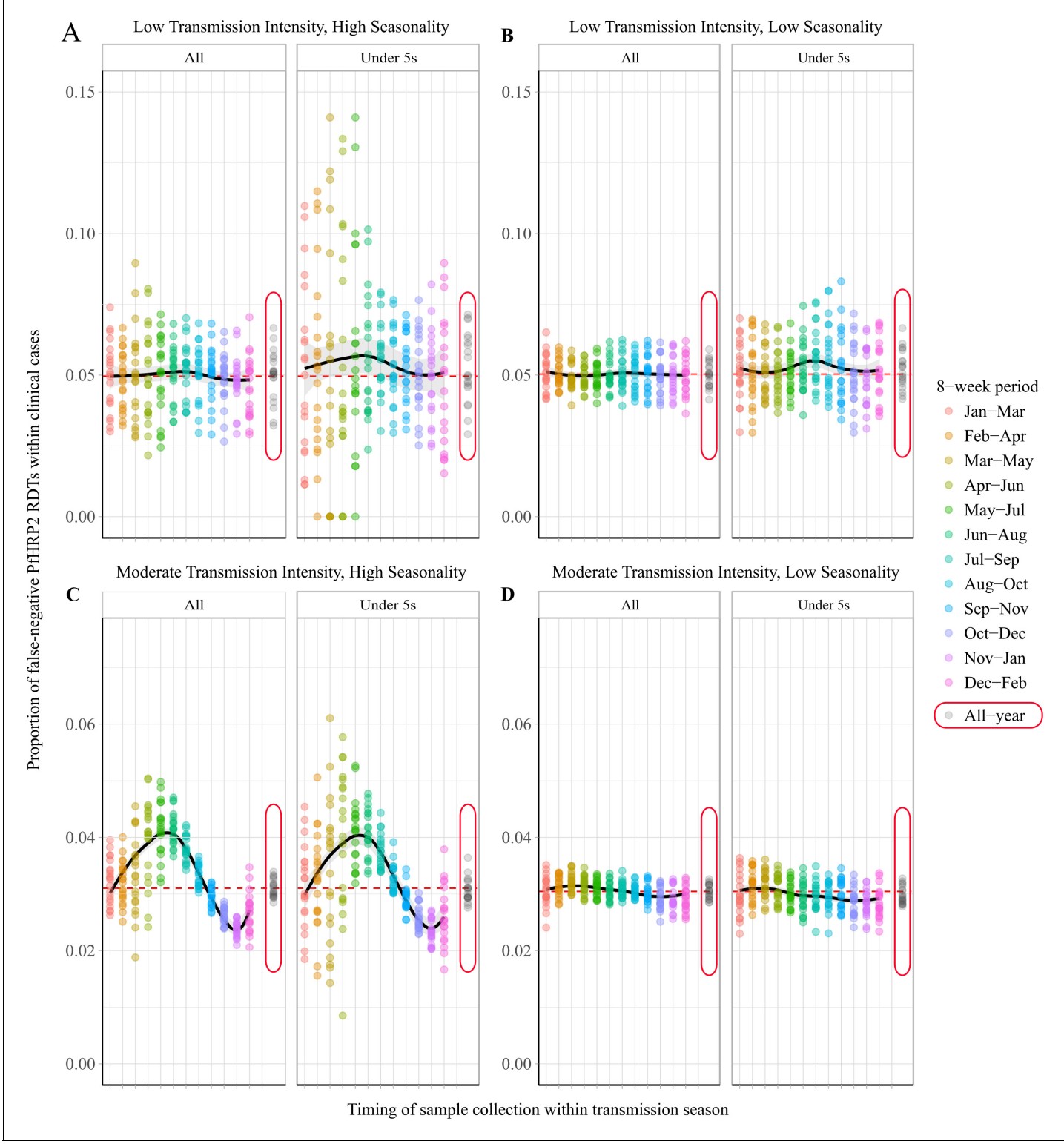

**Figure 2.** Observed proportion of false-negative PfHRP2 RDTs within clinical cases during 8-week intervals. Graphs show the proportion of clinical cases yielding false-negative PfHRP2 RDTs at 8-week intervals within a transmission season for both moderate (**C, D**) and low (**A, B**) transmission settings and high (**A, C**) and low (**B, D**) seasonality. In each panel, the observed proportion *pfhrp2*-negative clinical cases is shown for the whole population and within children aged under 5 years old. Ten stochastic realisations are represented by the points in each plot, with the mean relationship throughout the transmission shown in black with a locally weighted scatterplot smoothing regression (loess). The annual average proportion of false-negative RDTs

*Figure 2 continued*

due to *pfhrp2* gene deletions is shown with the horizontal dashed red line, and a sampling scheme that occurs throughout the year, with samples collected proportionally to clinical incidence, is shown with grey points circled in red.

DOI: https://doi.org/10.7554/eLife.40339.005

The following figure supplement is available for figure 2:

**Figure supplement 1.** Impact of a selective advantage for *pfhrp2*-deleted parasites on the observed proportion of false-negative PfHRP2 RDTs within clinical cases during 8-week intervals.

DOI: https://doi.org/10.7554/eLife.40339.006

observations from Dioro in the Ségou region of Mali, where in 2012 more than 80% of smear-positive individuals had false-negative RDTs when collected at the end of the dry season (***Koita et al., 2013***). The proportion of false-negative RDTs then rapidly decreased to 20% within 3–4 weeks after the start of the rainy season. It is, however, likely that a proportion of these false-negative RDTs were due to the increased observation of lower parasitaemia at lower transmission intensities such as at the end of the dry season (***Okell et al., 2012***). In addition, findings from Eritrea also support our model-predicted outcomes. Eritrea is a region with lower malaria prevalence compared to the Ségou region of Mali. The resultant decrease in transmission intensity is likely to result in an increased proportion of monoclonal infections throughout the transmission season. Consequently, we would predict less variability in the number of false-negative RDTs due to *pfhrp2* gene deletions at any given period within a transmission season. We also expect the observed prevalence of *pfhrp2* deletions to be more stochastic due to the lower effective population size of the parasite. Indeed, infections due to *pfhrp2*-deleted parasites identified in Eritrea between November 2013 and November 2014 were not more likely to have occurred after periods of lower transmission intensity (p=0.56, n=144, pfhrp2 deletions at 9.7%) (***Menegon et al., 2017***).

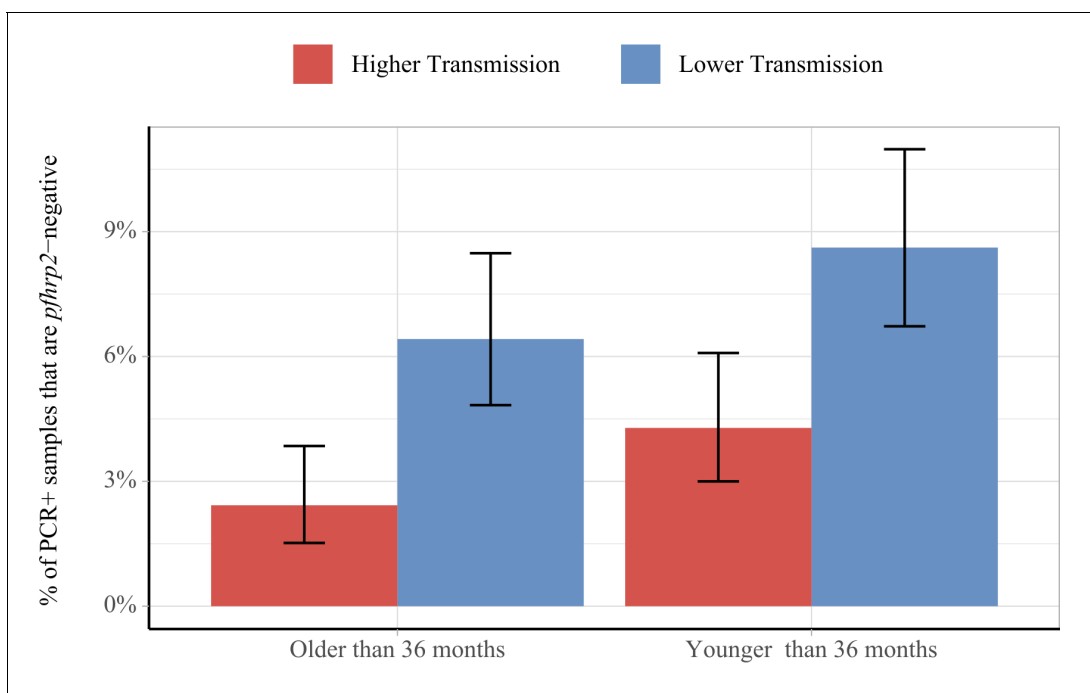

**Figure 3.** Impact of age and transmission intensity upon *pfhrp2* deletion in the Democratic Republic of the Congo (DRC), 2013–2014. Graphs show the percentage of PCR-positive *P. falciparum* samples taken from children under the age of 5 years from the 2013–2014 Demographic and Health Survey in DRC that are *pfhrp2*-negative. Children who are younger than the median age in the 2752 samples are grouped within the younger category. In addition, samples are classified as lower transmission if the incidence of malaria in the month prior to sample collection is lower than the median clinical incidence. The 95% binomial confidence intervals are indicated with the vertical error bars.

DOI: https://doi.org/10.7554/eLife.40339.007

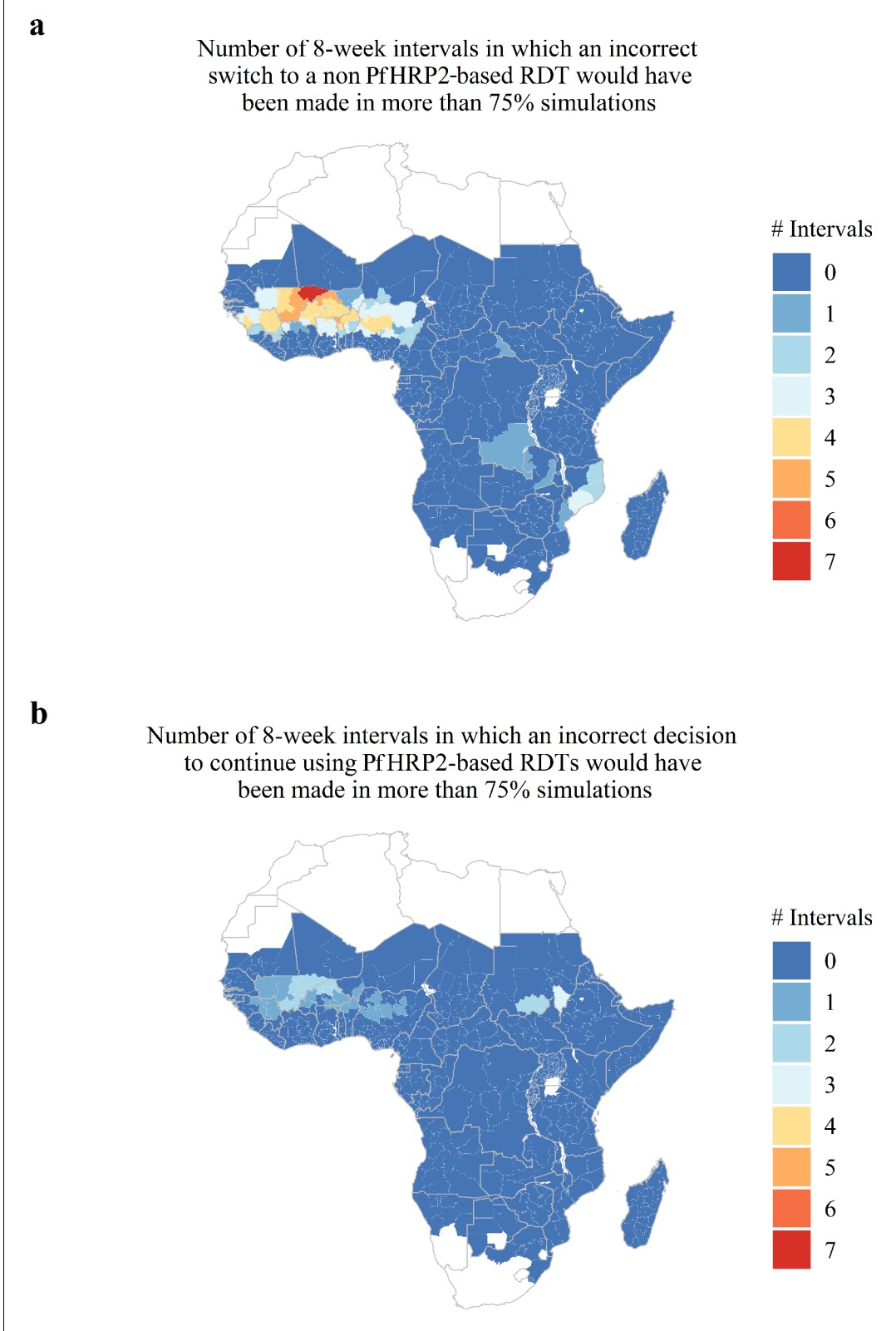

**Figure 4.** Predicted areas with the potential for collected estimates of the proportion of false-negative PfHRP2 RDTs due to *pfhrp2* deletions to be unrepresentative of the annual average. The maps show (A) the number of 8-week intervals at which an administrative region would prematurely swap to a non PfHRP2-based RDT due to overestimating the proportion of false-negative PfHRP2 RDTs due to *pfhrp2* gene deletions in more than 75% of simulations. In (A) the opposing trend is shown, with the number of 8-week intervals at which an administrative region would prematurely continue to

*Figure 4 continued*

use PfHRP2-based RDTs due to underestimating the proportion of false-negative PfHRP2 RDTs due to *pfhrp2* gene deletions in more than 75% of simulations.

DOI: https://doi.org/10.7554/eLife.40339.008

Similar to the original publication (*Watson et al., 2017*), there are a number of modelling assumptions in this study. Firstly, there are modelling uncertainties when predicting the dynamics of false-negative RDTs due to *pfhrp2*-deleted parasites. To account for this uncertainty in this analysis, we have controlled for the drivers characterised in our earlier study by assuming there was no selective advantage associated with *pfhrp2*-deleted parasites and recording the number of individuals who would have been *pfhrp2*-negative and subsequently misdiagnosed. The absence of a selective advantage in this way enabled the frequency of *pfhrp2* deletions to remain constant, which ensured that any observed dynamics in the estimates of false-negative RDTs due to *pfhrp2* deletions were due to the seasonality of transmission and not due to an increase in the population frequency of *pfhrp2* deletions. However, we are aware that there is likely a selective advantage for *pfhrp2* deleted parasites and subsequently we repeated the analyses with the selective advantage included. In these simulations, we predicted a substantial increase in the frequency of *pfhrp2* gene deletions (*Figure 1—figure supplement 2Q-T*), however clear seasonal dynamics, with an increased proportion of false-negative RDTs due to *pfhrp2* deletions at the beginning of the transmission season, were still observed (*Figure 2—figure supplement 1C*). However, the observed dynamics were less clear in settings with the greatest increase in the frequency of *pfhrp2* deletions (*Figure 2—figure supplement 1B*).

Secondly, we assessed the potential for a region to yield unrepresentative estimates of the proportion of false-negative RDTs due to *pfhrp2* deletions through comparisons to the annual average proportion. This decision reflected firstly the monitoring period defined in the WHO technical guidance, with follow-up studies recommended after two years if the 95% CI for the proportion of *P. falciparum* cases with false-negative HRP2 RDT results due to *pfhrp2/3* deletions is less than 5%, or one year if it does include 5%. It also reflected our modelling assumption that the population frequency of *pfhrp2* deletions is not increasing over time. However, in simulations in which a selective advantage to *pfhrp2* deleted parasites was included, a comparison to the annual average proportion is less suitable. For example, in *Figure 2—figure supplement 1B*, because we started our simulations in January the optimum sampling interval is simply the interval in the middle of the year, reflecting the constant increase in *pfhrp2* deleted parasites. In these scenarios, it could be argued that the correct comparison would be to the average proportion of false-negative RDTs due to *pfhrp2/3* gene deletions in the year after sampling, which reflects how many cases could be misdiagnosed between sampling rounds. Unfortunately, this comparison is difficult without knowing how the proportion of false-negative RDTs due to *pfhrp2/3* gene deletions will change over time. However, we believe that it is more important to focus on the assumption that the strength of selection is negligible (see *Figure 5*). Our rationale for this is that it is only in areas with a low selective pressure, for which the frequency of *pfhrp2/3* deletions is constant over time, that one could repeatedly make an incorrect decision with regards to whether to switch RDT (*Figure 5A*). In areas with a selective pressure, it is still possible to incorrectly estimate the annual average for the following year; however, the presence of the selective pressure is likely to cause any decision made to be simply premature as the frequency of *pfhrp2/3* deletions and subsequently false-negative PfHRP2 RDTs will increase over time (*Figure 5B*).

Lastly, it is important to note again that the true strength of selection is unknown. The precise strength of selection is dependent on a number of factors such as the magnitude of any fitness costs associated with *pfhrp2* deletion, the degree to which microscopy-based diagnosis is used, the level of non-adherence to RDT results, the treatment coverage and the prevalence of malaria in the region considered. Consequently, our results should not be interpreted as precise predictions of how unrepresentative future samples may be. They should instead be used to support surveillance efforts and to reinforce the need for longitudinal measures conducted at the same point within a transmission season. In addition, we recommend that if possible, sample collection in highly seasonal regions should not occur at the beginning of the transmission season, as this is predicted to lead to

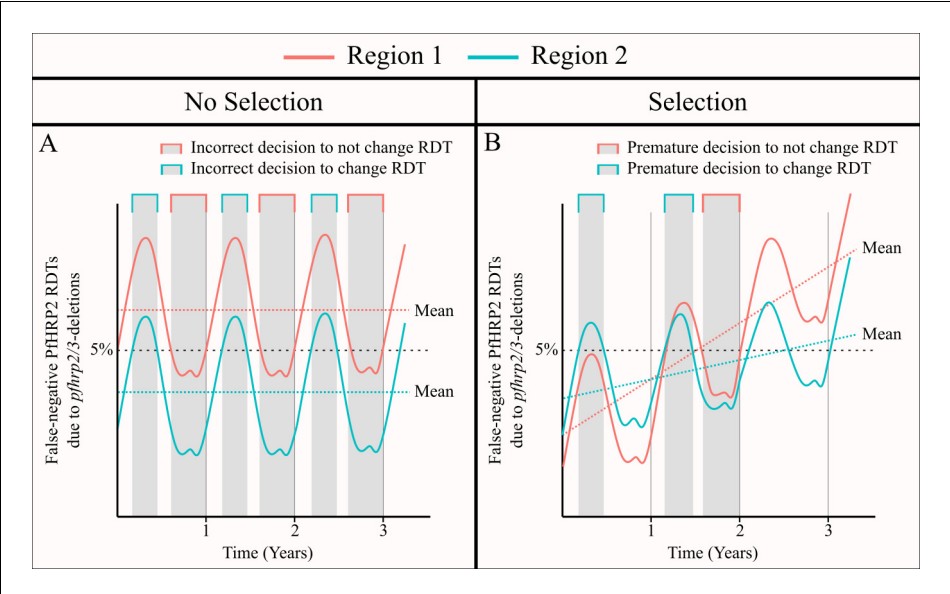

**Figure 5.** The impact of an assumed selective pressure for *pfhrp2/3*-deleted parasites on the decision to switch RDT. The graphs show two hypothetical scenarios with two different regions shown in red and blue for each region. In (**A**) there are strong seasonal dynamics but no selective pressure. The absence of a selective pressure causes that the mean proportion of false-negative RDTs due to *pfhrp2/3* deletions over a 1 year period to be constant and is shown with a horizontal dashed line. Consequently, there are time periods in which an incorrect decision to switch RDT could be made for the region in blue, and an incorrect decision to not switch RDT could be made for the region in red. In (**B**), there are both seasonal dynamics and a selective pressure, which results in an increasing annual mean proportion of false-negative RDTs due to *pfhrp2/3* deletions over time. As in (**A**), there are periods in which the observed proportion of false-negative RDTs due to *pfhrp2/3* deletions is both higher and lower than the rolling mean shown. However, decisions made in these periods are premature rather than definitively incorrect as the selection pressure would eventually cause the proportion to be greater than 5%.
DOI: https://doi.org/10.7554/eLife.40339.009

premature decisions to switch RDT irrespective of the strength of selection. It will, however, be possible after the samples have been collected to estimate the likely frequency of *pfhrp2* gene deletions by incorporating estimates of the multiplicity of infection within the sampled population. This frequency could then be used to estimate how the proportion of false-negative RDT results due to *pfhrp2* deletions could increase in response to decreases in the prevalence of malaria.

In summary, our extended model predicts that highly seasonal dynamics in malaria transmission intensity will cause comparable dynamics in the observed proportion of false-negative RDT results due to *pfhrp2* gene deletions. The observed proportion of false-negative RDTs due to *pfhrp2* deletions is higher when monoclonal infections are more prevalent, with the highest prevalence observed when sampling at the start of the rainy season as individuals are less likely to already be infected. Similarly, the observed proportion of false-negative RDTs due to *pfhrp2* deletions is higher in younger individuals who have lower clinical immunity, as they are more likely to present with clinical symptoms after their first infection event. As the rainy season progresses, individuals are more likely to be superinfected and acquire wild-type parasites, resulting in positive PfHRP2-based RDT results and a decrease in the observed proportion of false-negative RDTs due to *pfhrp2* deletions. In response to these dynamics, it may be sensible for national malaria control programmes conducting surveillance for *pfhrp2/3* deletions to choose a sampling interval towards the end of the transmission season, which is predicted to be most representative of the annual average proportion of false-negative RDTs due to *pfhrp2* deletions. To support surveillance efforts, we have published an online database detailing the optimum sampling interval as well as the fluctuations throughout the transmission season for each administrative region.

## Materials and methods

### Extensions to the *P. falciparum* transmission model

In our previous publication (*Watson et al., 2017*), we presented an extended version of an individual-based model of malaria transmission to characterise the key drivers of *pfhrp2* deletion selection; however, it did not capture seasonality. To address this, we incorporated seasonal variation in malaria transmission intensity through the inclusion of seasonal curves fitted to daily rainfall data available from the US Climate Prediction Center (*National Weather Service Climate Prediction Center, 2010*). Rainfall data was available at a $10 \times 10$ km spatial resolution from 2002 to 2009, with data missing for only two days. The data was subsequently aggregated to a series of 64 points per year, before Fourier analysis was conducted to capture the seasonal dynamics within this time period (*Cairns et al., 2012*). The first three frequencies of the resultant Fourier transformed data were used to generate a normalised seasonal curve. This inclusion alters the rate at which new adult mosquitoes are born, with the differential equation governing the susceptible adult stage of the mosquito population now given by:

$$\frac{dS_M}{dt} = \theta(t)\mu_M M_v - \mu_M S_M - \Lambda_M S_M$$

where $\mu_M$ is the daily death rate of adult mosquitoes, $M_v$ is the total mosquito population, that is $S_M + E_M + I_M$, $\Lambda_M$ is the force of infection on the mosquito population and $\theta(t)$ is the normalised seasonal curve, with a period equal to 365 days. The rest of the model equations remain the same as in our original study (*Watson et al., 2017*).

All extensions to the previous model code have been made using the R language (RRID:SCR_001905) (*R Development Core Team, 2016*) and are available through an open source MIT licence at https://github.com/OJWatson/hrp2malaRia (*Watson, 2019*; copy archived at https://github.com/eLifeProduction/hrp2malaRia_2019). In addition, these extensions have been included in the pseudo-code description of the model (*Supplementary file 1*).

### Characterising the impact of seasonal transmission intensities upon the proportion of false-negative RDTs due to *pfhrp2* gene deletions

The impact of seasonality was examined by recording the proportion of clinical incidence that would have been misdiagnosed due to *pfhrp2* gene deletions across the year. This proportion was summarised at 12 8-week intervals, that is January – March, February – April, December – February. This proportion was recorded in both high and low seasonality settings, characterised by a Markham Seasonality Index = 80% and 10%, respectively (*Cairns et al., 2015*). These settings were examined at both low and moderate transmission intensity (EIR = 1 and 10 respectively), with the starting proportion of *pfhrp2*-deleted parasites in the whole population set equal to 6% in agreement with previous observations of *pfhrp2* gene deletions in the DRC (*Watson et al., 2017*) The proportion of symptomatic cases seeking treatment was assumed to be 40% ($f_T$pfhrp2 = 0.4). In all simulations, 10 stochastic realisations of 100,000 individuals were simulated for 60 years to reach equilibrium first, before setting the frequency of *pfhrp2* deletions. Initially, we assumed there was no assumed fitness cost or selective advantage associated with *pfhrp2* gene deletion. This was modelled by assuming that individuals who are only infected with parasites with *pfhrp2* gene deletions will still be treated. This decision allowed us to control for selection within our investigation by ensuring that the changes observed in the observation of PfHRP2-negative clinical cases are only due to seasonal variation in transmission intensity, and not due to an increase in the frequency of *pfhrp2* gene deletions due to the selective advantage by evading diagnosis. As a result, when reporting the proportion of clinical cases that were misdiagnosed resulting from a false-negative PfHPR2-negative RDT we are reporting the proportion of cases that are infected with only *pfhrp2*-deleted parasites, that is individuals who would have been *pfhrp2*-negative and subsequently misdiagnosed. We also assume that 25% of individuals who are only infected with *pfhrp2*-deleted parasites will still be *pfhrp2*-positive due to the cross reactivity of PfHRP3 epitopes causing a positive PfHRP2-based RDT result (*Baker et al., 2005*).

Model predictions were subsequently compared to data collected from the Democratic Republic of Congo as part of their 2013–2014 Demographic and Health Survey (DHS). In overview, 7137 blood samples were collected from children under the age of 5 years old, which yielded 2752

children diagnosed with *P. falciparum* infection by real-time PCR targeting the lactate dehydrogenase (*pfldh*) gene. The RDT barcodes for the 2752 samples were identified and matched to the DHS survey to identify both the age of the children and the date of sample collection. The collection date was used to predict the mean clinical incidence from the previous 30 days for each sample. This was estimated using the deterministic implementation of our model fitted to the observed PCR prevalence of malaria from the DRC DHS 2013–2014 survey (*Meshnick et al., 2013*), incorporating the seasonality and treatment coverage for each province. Children who were younger than the median age in the 2752 samples were grouped within a younger category. In addition, samples were classified as lower transmission if the clinical incidence of malaria in the month prior to sample collection was lower than the median clinical incidence. The counts of *pfhrp2*-negative samples within each group were subsequently compared using the Pearson chi-squared test with Rao-Scott corrections to account for the hierarchal survey design implemented within DHS surveys (*Jnk and Scott, 1984*). Pearson chi-squared tests were used in a similar analysis that was conducted using samples collected from the Gash-Barka and Debug regions in Eritrea between 2013 and 2014, for which the dates of sample collection were made available to us (*Menegon et al., 2017*).

Finally, the seasonal profiles for 598 first-level administrative regions across sub-Saharan Africa were used to characterise the potential for estimates of the proportion of false-negative PfHRP2 RDTs due to *pfhrp2* gene deletions to be unrepresentative of the annual average. For each region, 100 simulation repetitions were conducted for 60 years to reach equilibrium first before fitting the frequency of *pfhrp2* gene deletions in each simulation such that the annual average proportion of false-negative RDT results due to *pfhrp2* deletions is equal to 5%. Each repetition was subsequently simulated for two further years, with 7300 individuals seeking treatment sampled from each 8-week interval. This number approximates the recommended sample size within the WHO protocol for *pfhrp2* deletion prevalence at $5 \pm 0.5\%$. For each sample, the proportion of false-negative PfHRP2-based RDTs due to *pfhrp2* gene deletions was recorded. For each sample, a binomial confidence interval was calculated and the resultant percentage of intervals that did not include the annual prevalence of 5% was calculated. For each region, the number of 8-week intervals for which a premature decision to either swap from a PfHRP2-based RDT or continue using a PfHRP2-based RDT was made in more than 75% of simulations was recorded and mapped. The raw results of this analysis were subsequently used to create a database that details the optimum sampling intervals for estimating the annual proportion of false-negative RDT results due to *pfhrp2* deletions.

## Acknowledgements

We thank Dr Michela Menegon for sharing the sample dates for the 2013–2014 study in Eritrea and the administrators and participants of the Demographic and Health Surveys.

## Additional information

### Funding

| Funder | Grant reference number | Author |
|---|---|---|
| Wellcome Trust | 109312/Z/15/Z | Oliver John Watson |
| Medical Research Council | MR/N01507X/1 | Robert Verity |
| Department for International Development | | Azra C Ghani |
| Medical Research Council | | Tini Garske |
| National Institute of Allergy and Infectious Diseases | R01AI132547 | Steven R Meshnick |
| American Society for Tropical Medicine and Hygiene | Burroughs Wellcome Fund-ASTMH Postdoctoral Fellowship in Tropical Infectious Diseases | Jonathan B Parr |

| Burroughs Wellcome Fund | Burroughs Wellcome Fund-ASTMH Postdoctoral Fellowship in Tropical Infectious Diseases | Jonathan B Parr |
|---|---|---|
| Imperial College London | | Hannah C Slater |

The funders had no role in study design, data collection and interpretation, or the decision to submit the work for publication.

## Author contributions

Oliver John Watson, Conceptualization, Data curation, Software, Formal analysis, Investigation, Methodology, Writing—original draft, Project administration, Writing—review and editing; Robert Verity, Formal analysis, Supervision, Visualization, Methodology, Writing—review and editing; Azra C Ghani, Jane Cunningham, Supervision, Methodology, Writing—review and editing; Tini Garske, Data curation, Software, Methodology, Writing—review and editing; Antoinette Tshefu, Melchior K Mwandagalirwa, Steven R Meshnick, Resources, Formal analysis, Writing—review and editing; Jonathan B Parr, Conceptualization, Data curation, Formal analysis, Supervision, Methodology, Writing—review and editing; Hannah C Slater, Conceptualization, Formal analysis, Supervision, Investigation, Visualization, Methodology, Writing—review and editing

## Author ORCIDs

Oliver John Watson  https://orcid.org/0000-0003-2374-0741

## Decision letter and Author response

Decision letter https://doi.org/10.7554/eLife.40339.016
Author response https://doi.org/10.7554/eLife.40339.017

# Additional files

## Supplementary files

• Supplementary file 1. Simulation model pseudocode. Mathematical style pseudocode description of the simulation model.
DOI: https://doi.org/10.7554/eLife.40339.010

• Transparent reporting form
DOI: https://doi.org/10.7554/eLife.40339.011

## Data availability

All data generated are provided within the online database, hosted through a shiny application at https://ojwatson.shinyapps.io/seasonal_hrp2/. The raw data for the application is available within the GitHub repository at https://github.com/OJWatson/hrp2malaRia (copy archived at https://github.com/eLifeProduction/hrp2malaRia_2019).

The following previously published dataset was used:

| Author(s) | Year | Dataset title | Dataset URL | Database and Identifier |
|---|---|---|---|---|
| Bhatt S, Weiss DJ, Cameron E, Bisanzio D | 2015 | PfPR2-10 in Africa 2000-2015 | https://map.ox.ac.uk/explorer/#/explorer | Malaria Atlas Project, explorer |

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
