## [Decision Letter]

Thank you for submitting your article "The impact of seasonal variations in *Plasmodium falciparum* malaria transmission on the surveillance of *pfhrp2* gene deletions" for consideration by *eLife*. Your article has been reviewed by three peer reviewers, including Ben Cooper as the Reviewing Editor and Reviewer #1, and the evaluation has been overseen by Prabhat Jha as the Senior Editor. The following individuals involved in review of your submission have agreed to reveal their identity: Elena Gómez-Díaz (Reviewer #2); Penelope Anne Lynch (Reviewer #3).

The reviewers have discussed the reviews with one another and the Reviewing Editor has drafted this decision to help you prepare a revised submission.

Summary:

This research represents a research advance which builds on a previous paper by the same group which considered the selection pressure exerted by the widespread use of rapid diagnostic tests for malaria in sub-Saharan Africa for deletions to the *pfhrp2* gene, which can lead to false negative test results. Subsequent to the previous publication, the World Health Organization has produced guidance for the investigation of suspected false-negative diagnostic test results due to *pfhrp2/3* deletions. However, this guidance says nothing about the recommended timing of such investigations. The current work uses an extension of the model-based analysis to show that seasonal variation in malaria transmission can lead to substantial biases in estimates of the prevalence of *pfhrp2/3* deletions, leading to poor choices of rapid diagnostic test. The risk of this sampling bias is mapped by region, and optimum sampling intervals are proposed.

Essential revisions:

All reviewers thought that the work was important and conducted to a high standard and should be published if some essential revisions are made. These revisions are needed primarily to improve the clarity of the work and in some cases to extend the Discussion to consider other factors that might be important (see individual reviews below). Comments marked with a * below should be considered discretionary revisions. In particular, though non-essential, it was felt that an additional figure might help to clarify the relationship between monoclonal/multiclonal infections and *pfhrp2* deletions prevalence and selection. An important part of these clarifications is provision of pseudo code for the revised model, just to document exactly how the updated DDEs shown in the manuscript are incorporated into the simulation model.

Reviewer #1:

The seems to be a useful research advance that addresses an important policy question using a model described in a previous *eLife* paper. The work is well-motivated and clearly described.

Reviewer #2:

This is an a research advance upon a previous study Watson et al., 2017.

In the previous article, authors modeled the potential for RDT-led diagnosis to drive selection of *pfhrp2*-deleted parasites. In the present work, authors extend the model so it now considers the impact of transmission intensity and seasonality on the prevalence of *pfhrp2* gene deletions. They found that regions with low transmissibility and high seasonality are those with higher number of false negatives (higher prevalence of *pfhrp2* deletions). They also show that this bias is stronger in young children.

The article is clearly written, the figures are very illustrative, and the new data support the conclusions. The new findings are significant. The data provided represents an important resource for the community.

- The extended analysis focus on seasonality and transmission intensity. I wonder about other possible causes of RDTs misdiagnosis. For example, the work seem to focus only on the clinical cases. What is the dynamics expected for *pfhrp2* deletions in the asymptomatic? This is important because asymptomatic malaria significantly impacts transmission dynamics and asymptomatic infections show seasonality.

- The study model *pfhrp2* deletions but no consideration is made about the effect of the type of treatment driving selection. There might be a temporal and spatial variability at this regards that has not been considered?

*- The link between transmission intensity and multiplicity of infections is clear. However, I find confusing the relationship between monoclonal/multiclonal infections and *pfhrp2* deletions prevalence and selection. I think this should be elaborated further and possibly modeled?

- Previous studies indicated that PfHRP3 may play a role in the performance of PfHRP2-based RDTs. Do authors have data on *pfhrp3*? Apart of *pfhrp2* deletions, could other sequence differences contribute to lower sensitivity of RDTs?

Reviewer #3:

This paper provides novel insights into an issue of practical public health importance. The results are interesting, and deserve to be disseminated and understood. In order to achieve this fully, the paper would benefit from greater clarity in some areas. Elements of the story which are perhaps viewed as self-evident by the authors may not be self-evident to readers, and are key to interpreting the paper and its results.

This paper adds seasonal variation to an individual-based model simulating prevalence of *pfhpr2*-del strains and false negative results in a population over time,. I have not attempted to check the original model, but the amendment shown in the current paper seems correct. Could the authors provide an updated version of the pseudo-code documentation reflecting the updates?

WHO guidelines recommend a transition from HPR2-based RDTs to alternatives when the prevalence of false-negatives due to *pfhpr2* deletion exceeds 5%, and specify survey protocols to test for this. This paper focusses on potential biases in the survey results arising from variation due to effects of seasonality and transmission intensity. Since it is central to the paper's premise, a brief explanation of the WHO survey protocol is needed, with an explicit explanation of the links between the simulation outputs and the values measured in the protocol. I think the relevant values are all present in the paper, but their meaning and relationships could be more clearly explained.

Can the authors clarify the basis on which the 5% threshold value was selected by WHO? The bias discussed in the paper may have different implications depending on whether the key comparator is the underlying prevalence of *pfhrp2/3* gene deletions or the annual average proportion of *pfhrp2/3*-del false negatives. Is there any potential to add some discussion about the implications of this study for the WHO threshold value, for example whether specific values could be specified for particular seasonality and transmission-intensity contexts?

The text regarding assumptions about selection and fitness (copied below) is confusing. False negative RDTs and consequent treatment choices reflected in the model will inherently exert selection, which seems to conflict with statements in the text.

'Additionally, there was no assumed fitness cost or selective advantage associated with *pfhrp2* gene deletion, i.e. individuals who are only infected with parasites with *pfhrp2* gene deletions are assumed to yield a false-negative RDT result. This decision allowed us to control for selection within our investigation. This ensures that the dynamics observed are only due to seasonal variation in transmission intensity, and not due to an increase in the frequency of *pfhrp2* gene deletions due to a selective advantage by evading diagnosis. As a result, when reporting individuals who are *pfhrp2*-negative we assume that 25% of individuals who are only infected with *pfhrp2*-deleted parasites will still be *pfhrp2*-positive due to the cross reactivity of PfHRP3 epitopes causing a positive PfHRP2-based RDT result.'

[Editors' note: further revisions were requested prior to acceptance, as described below.]

Thank you for resubmitting your work entitled "Impact of seasonal variations in *Plasmodium falciparum* malaria transmission on the surveillance of *pfhrp2* gene deletions" for further consideration at *eLife*. Your revised article has been favorably evaluated by Prabhat Jha as the Senior Editor, a Reviewing Editor, and two reviewers.

The manuscript has been improved but there are some remaining issues that need to be addressed before acceptance, as outlined below:

As you know, there was some confusion in this paper, as the original submission indicated that model did account for selection for *pfhrp2* mutants, but the subsequent correspondence indicated that the model didn't.

While we understand that there is some value in considering the situation where the frequency of *pfhrp2* deletions is not affected by selective forces (i.e. delayed treatment), clearly selective forces are likely to be acting in most settings and following consultation the consensus was that completely removing this real-world effect from the model was hard to justify. Therefore in addition to the analysis that has been done, the authors should add additional work where they do what they originally said they had done i.e. including a model where there is a selective advantage for *pfhrp2* deletion changes/mutants as originally indicated.

The authors appear to be assuming that the intended meaning of the 5% threshold is the average proportion of HRP2 RDT results for patients infected with *P. falciparum* which are false negatives caused by *pfhrp2/3* gene deletions, during a given year. The Discussion and conclusion then focus on differences between the prevalence of false negatives at specific timepoints during a year vs the average prevalence value over the year. It would hugely improve the clarity of the paper to state this assumption explicitly and early in the text. It is also necessary to demonstrate using information in the WHO documentation that this is in fact the intended definition of the WHO 5% threshold. Without unequivocal evidence that this is the precise meaning of the threshold value intended by the WHO, then the use of terms such as 'bias', 'overestimate', 'underestimate' etc. is unjustified throughout.

If it is not clear that the threshold is defined as an annual average, then the paper's message needs to change slightly. By indicating the extent to which the prevalence of false negatives can vary seasonally, even when the prevalence of gene deletions is constant, the results presented here indicate that a conscious choice about this aspect of the definition is very important. Should the threshold represent the acceptable maximum prevalence of false positives, or should it be the annual average. In either case, the results can inform strategies for applying the protocol in ways most likely to identify the required value.

Because of the extensive nature of the requested revisions and clarifications which cannot easily be summarized, more extensive comments from both reviewers are appended below. All substantive points should be addressed satisfactorily as we are unable to extend the review process beyond this next revision.

Reviewer #2:

The manuscript has improved and authors addressed most of my comments satisfactorily. I have however a few additional comments on the revised manuscript and rebuttal letter which I feel would require additional clarification.

- Exceptionally unhelpfully, the use of "false-negative" should be "positive" here. We carried out all our simulations with the assumption that individuals who are only infected with *pfhrp2* gene deleted parasites will still be treated. As such, the gene deleted parasites behave exactly the same as the wild type parasites.

I am afraid I don't fully follow this reasoning. My understanding is that the motivation of the study was that *pfhrp2* gene deleted parasites could be indeed misdiagnosed and so simulations should treat them as false negatives (Introduction, second paragraph). If simulations threat those as positive, how could the model effectively estimate the rate of misdiagnosis and the seasonality in such estimate? May I have missed something?

Besides, I don't think that the reviewers have actually addressed the real concern that came with their original consideration that false negative RTD *pfhrp2* deleted parasites would allow them to control for selection.

- Related to the same issue above, and in response to my comment, authors replied:

"We do not include a selective advantage to *pfhrp2* gene deletion (apologies again for the error mentioned at the beginning of our response) and so we would not expect there to see a temporal variability in the selection pressure. If we did consider this then there would definitely be a temporal element, with the increase in the absolute number of people who seek treatment (we assume a constant proportion of people with a malarial fever seek treatment) during periods of higher transmission causing an increase in the prevalence of the *pfhrp2* gene deletion. It was because of this reason that we decided not to model selection, so that we could exclude this effect of selection and be more confident that the dynamics seen are due to the fluctuations in individuals being only infected with *pfhrp2* deleted parasites."

The selective advantage comes with *pfhrp2* gene deletion individuals being misdiagnosed and not getting treatment. If you consider those as positive you remove selection but this is not reflecting any more the reality of the situation.

- About the relationship between monoclonal/multiclonal infections and *pfhrp2* deletions prevalence and selection.

I thank the authors about including a supplementary figure, but could it be possible to clarify further the relationships in the text?, saying that the relationship is unclear is not of much help.

- About my comment "The regions identified were areas with both a low prevalence of malaria and a high frequency of people seeking…" Were these the only factors?"

To which authors responded "These were the only factors we looked at within our modelling study".

I don't find this reply satisfactory. I know they modelled only those, but my comment was more a recommendation so it is acknowledged somewhere in the Introduction or the Discussion whether they could be other factors that have not been considered and have been shown or suggested to influence the misdiagnoses.

Reviewer #3:

The author's clarifications make sense and are helpful. However, my improved understanding of the authors' intentions and the results and conclusions presented in the paper has generated some additional questions and comments. I still feel that the paper would benefit from greater clarity.

My understanding is that the key values being considered are;

1) The proportion of HRP2 RDT results for patients infected with *P. falciparum* which are false negatives caused by *pfhrp2/3* gene deletions at a given timepoint.

2) The average proportion of HRP2 RDT results for patients infected with *P. falciparum* which are false negatives caused by *pfhrp2/3* gene deletions, during a given year.

3) The proportion of *P. falciparum* parasites in a given region which have *pfhrp2/3* gene deletions.

4) The 5% threshold in the WHO guidelines.

It would be incredibly helpful if the authors could provide a precise definition for this, as the various wordings I have found so far in the WHO protocol and information note are open to interpretation regarding whether the 5% is intended to represent: a) The proportion of HRP2 RDT results for patients infected with *P. falciparum* which are false negatives caused by *pfhrp2/3* gene deletions; or b) The proportion of *P. falciparum* parasites in a given region which have *pfhrp2/3* gene deletions.

Part of a full definition for this value is the assumed timing. A quick review of the WHO documentation does not immediately yield any specific information about assumed timings, an absence which would be consistent with an assumption that the rate is effectively constant through a season, or might equally mean that the relevant value is that at the time of sampling.

In the paper, the authors appear to be assuming that the intended meaning of the 5% threshold is the average proportion of HRP2 RDT results for patients infected with *P. falciparum* which are false negatives caused by *pfhrp2/3* gene deletions, during a given year (item 2 in the list above). The Discussion and conclusion then focus on differences between the prevalence of false negatives at specific timepoints during a year vs the average prevalence value over the year. It would hugely improve the clarity of the paper to state this assumption explicitly and early in the text. It is also necessary to demonstrate using information in the WHO documentation that this is in fact the intended definition of the WHO 5% threshold. Without unequivocal evidence that this is the precise meaning of the threshold value intended by the WHO, then the use of terms such as 'bias', 'overestimate', 'underestimate' etc. is unjustified throughout.

If it is not clear that the threshold is defined as an annual average, then the paper's message needs to change slightly. By indicating the extent to which the prevalence of false negatives can vary seasonally, even when the prevalence of gene deletions is constant, the results presented here indicate that a conscious choice about this aspect of the definition is very important. Should the threshold represent the acceptable maximum prevalence of false positives, or should it be the annual average. In either case, the results can inform strategies for applying the protocol in ways most likely to identify the required value.

There is also some confusion in the text between the prevalence of false positives results, and the prevalence of the gene deletion, with the text referring to change of RDT being triggered by an incorrect assessment of the prevalence of gene deletions (e.g. Introduction, fourth paragraph), suggesting that the authors may in fact be defining the threshold value as equal to value 3 in the list above.

These are key to the meaning and the implications of the work presented here, and clarity about what is being assumed or referred to is crucial to allow the text to tell its story clearly, and to make it easy to assess the consistency of that story. Confusing references to different prevalence values in the text should be reviewed and resolved wherever they arise throughout the text, including some specific instances detailed below.

Detailed comments:

Introduction, third and fourth paragraphs: In the third paragraph of the Introduction the authors give a definition of the WHO threshold value as being the prevalence of false negatives caused by *pfhrp2/3* gene deletions. However, in the fourth paragraph of the Introduction they suggest that incorrect assessment of the prevalence of *pfhrp2/3* gene deletions could drive the decision to switch to non HRP2 RDTs. Is there another mechanism in the WHO guideline in addition to the 5% false negatives threshold which would drive a change of policy based on gene deletion prevalence rather than false negative RTD prevalence?

'The protocol in this guidance details how to estimate the local prevalence of false-negative PfHRP2-based RDTs due to *pfhrp2/3* gene deletions and recommends that a national change to non PfHRP2-based RDTs be made if the estimated prevalence is above 5%.'

'the timing of the 8-week interval chosen within a transmission season could lead to bias in the sampled prevalence of *pfhrp2/3* gene deletions. An overestimation of the true prevalence of *pfhrp2/3* gene deletions could result in a switch to a less sensitive RDT'

Results, first paragraph and similar elsewhere in text: 'In a moderate transmission setting, a clear seasonal pattern is predicted (Figure 2C), with sampling at the beginning of the transmission seasons resulting in significant overestimation of the true proportion of false negative RDTs..'

'true' is not adequately defined to be used here in this way. It might legitimately be assumed to mean the population prevalence of false-negative RDTs at the time of sampling. What is meant here, I think, is that sampling at the beginning of the transmission season is expected to give a value higher than the true average value for the year.

Introduction, last paragraph, Figure 4 description and title, Results, last paragraph.

Introduction, last paragraph and figure description indicate that the values used to generate Figure 4 are the gene deletion prevalences

Results, last paragraph and implication of contents of plot indicate that the plot is based on prevalences of false negative values.

Results, first and last paragraph and Discussion, first paragraph and similar elsewhere in text – 'biased' and 'unbiased' are a mathematical terms with specific meanings and it is not clear that those meanings are correctly applied here and elsewhere in the text. It would be better to replace them with other terms unless the mathematical meaning is genuinely indicated.

Discussion, first and second paragraphs. These paragraphs both begin by describing the research presented in the manuscript as relating to estimates of prevalence of *pfhrp2* gene deletions. The remaining text all seems to actually describe the results regarding the prevalence of false positive HRP2 RDT results, but the first sentences mean that it all reads as discussion of the gene deletion prevalence.

'This research characterises the potential for surveillance in highly seasonal areas within sub-Saharan Africa to produce biased estimates of the prevalence of *pfhrp2* gene deletions. These findings highlight the impact of both the seasonal timing and…'

'Our modelling predicted that there would be increased observation of *pfhrp2* gene deletions after periods of lower transmission and within younger individuals…'

Discussion, first, third and fourth paragraphs. 'However, the true prevalence of parasites with a *pfhrp2* gene deletion in each administrative region is fundamentally unknown, and as such, our results should not be interpreted as predictions of the bias in future sampled estimates of *pfhrp2* deletion. They should instead be used to support surveillance efforts and to reinforce the need for longitudinal measures of *pfhrp2* gene deletions conducted at the same point with a transmission season.'

Is this compatible with the database mentioned in the Discussion? 'To support surveillance efforts, we have published an online database detailing the optimum sampling interval as well as the sampling bias throughout the transmission season for each administrative region'

'The observed prevalence of *pfhrp2* deletions is higher when monoclonal infections are more prevalent, with the highest prevalence observed when sampling at the start of the rainy season as individuals are less likely to already be infected. Similarly, the observed prevalence of *pfhrp2* deletions is higher in younger individuals who have lower clinical immunity, as they are more likely to present with clinical symptoms after their first infection event.'

Should these two references be to prevalence of false positives rather than prevalence of *pfhrp2* deletions?

Discussion, last paragraph. This seems to be simply repeating contents of first paragraph of Discussion?

Subsection “Characterising the impact of seasonal transmission intensities upon *pfhrp2* deletion prevalence”, last paragraph. '…fitting the frequency of *pfhrp2* gene deletions in each simulation such that the true prevalence of false-negative RDT results due to *pfhrp2* deletions is equal to 5%.'

'.. percentage of intervals that did not include the true prevalence of 5% was calculated.'

'true' not adequately defined, should simply say '.. the average annual prevalence..' or similar.

Figure 1 legend. '..In I – L and M – P the proportion of clinical cases due to *pfhrp2*-negative parasites is shown for both the whole population and..'

Wording is confusing, does this mean cases infected only with *pfhrp2*-negative parasites?

'…the population allele frequency of *pfhrp2* gene deletions, which was set equal to 6% at the beginning of each simulation..'

Is the reason for or significance of the 6% value given anywhere?

'…10 simulation realisations are shown in each graph, with the mean shown with the thicker line. Lastly, the 5% threshold for switching RDT provided by the WHO is shown with the black line in plots I – P…'

I think the means are shown by the black line, and the 5% by the dashed horizontal line?

Figure 3 legend. Should '..age and seasonality..' be '..age and transmission intensity..'?

Figure 4 legend, description and title. '..*pfhrp2* deletion..' should be '..false-negative *pfhrp2* RDTs?..'

Should also be revised as necessary to reflect assumed exact definition of threshold value.

pseudo codesecond line 048

'// Loop through every day in simulation and calculate the seasonal curve for that day

045 FOR day: = 1 TO t_max // t_max is total simulation time in days

046 theta[day]:= Fourier_average +first_cosine_term * cos(2*pi*day/365) +second_cosine_term * cos(2*2*pi*day/365) +third_cosine_term * cos(3*2*pi*day/365) +first_sine_term * sin(2*pi*day/365) +second_sine_term * sin(2*2*pi*day/365) +third_sine_term * sin(3*2*pi*day/365))

047 ENDFOR

// Loop through every day in simulation and normalise seasonal curve for that day

048 FOR day: = 1 TO t_max // t_max is total simulation time in days

048 theta[day]: = theta [day] / mean(theta [1 TO 365) // normalise theta with first 365 days of theta

049 IF ([day] < 0.001) // with only 1st 3 terms of Fourier used we need to check for <0

050 [day]: = 0.001

051 ENDIF

052 ENDFOR

I'm assuming this is just a problem with the pseudo code, not the actual code, but that should be checked and confirmed. It seems that in the normalisation loop, the sum of theta values by which θ(n) is divided will use the normalised rather than original values for all θ(<n).

Could the authors please review the pseudo code for consistency with the actual code?

---

## [Author Response]

Essential revisions:All reviewers thought that the work was important and conducted to a high standard and should be published if some essential revisions are made. These revisions are needed primarily to improve the clarity of the work and in some cases to extend the Discussion to consider other factors that might be important (see individual reviews below). Comments marked with a * below should be considered discretionary revisions. In particular, though non-essential, it was felt that an additional figure might help to clarify the relationship between monoclonal/multiclonal infections and pfhrp2 deletions prevalence and selection. An important part of these clarifications is provision of pseudo code for the revised model, just to document exactly how the updated DDEs shown in the manuscript are incorporated into the simulation model.

Thank you to all the reviewers for their thoughtful comments and kind words. We have responded to all the comments below, and we would like to make one clarification initially here as it was picked up by all three reviewers. There was an error in the following:

“Additionally, there was no assumed fitness cost or selective advantage associated with *pfhrp2* gene deletion, i.e. individuals who are only infected with parasites with *pfhrp2* gene deletions are assumed to yield a false-negative RDT result. This decision allowed us to control for selection within our investigation"

Exceptionally unhelpfully, the use of “false-negative” should be “positive” here. We carried out all our simulations with the assumption that individuals who are only infected with *pfhrp2* gene deleted parasites will still be treated. As such, the gene deleted parasites behave exactly the same as the wild type parasites.

The reason for simulating it this way was so that we could see the impact of seasonality on the appearance of individuals that are only infected with *pfhrp2*-deleted parasites, i.e. those that would be misdiagnosed. We knew from the original study that at low transmission there is a strong selection pressure in favour of *pfhrp2*-deleted parasite. Subsequently, over the timespan we consider in the simulations it would have been likely that the frequency of *pfhrp2* gene deletions would have increased. This would have made it less clear how the dynamics of individuals only infected with *pfhrp2* deleted parasites changes throughout a transmission season, if the prevalence of the gene deletion is substantially higher at the end of a transmission season. Our apologies for the confusion and thank you for picking up on it. This section now reads as follows:

“Additionally, there was no assumed fitness cost or selective advantage associated with *pfhrp2* gene deletion. This was modelled by assuming that individuals who are only infected with parasites with *pfhrp2* gene deletions will still be treated. This decision allowed us to control for selection within our investigation by ensuring that the changes observed in the observation of PfHRP2-negative clinical cases are only due to seasonal variation in transmission intensity, and not due to an increase in the frequency of *pfhrp2* gene deletions due to the selective advantage by evading diagnosis.”

Reviewer #2:This is an a research advance upon a previous study Watson et al., 2017.In the previous article, authors modeled the potential for RDT-led diagnosis to drive selection of pfhrp2-deleted parasites. In the present work, authors extend the model so it now considers the impact of transmission intensity and seasonality on the prevalence of pfhrp2 gene deletions. They found that regions with low transmissibility and high seasonality are those with higher number of false negatives (higher prevalence of pfhrp2 deletions). They also show that this bias is stronger in young children.The article is clearly written, the figures are very illustrative, and the new data support the conclusions. The new findings are significant. The data provided represents an important resource for the community.- The extended analysis focus on seasonality and transmission intensity. I wonder about other possible causes of RDTs misdiagnosis. For example, the work seem to focus only on the clinical cases. What is the dynamics expected for pfhrp2 deletions in the asymptomatic? This is important because asymptomatic malaria significantly impacts transmission dynamics and asymptomatic infections show seasonality.

We agree that considering asymptomatic individuals is very important, especially as they are the major driver of onwards transmission. Additionally, understanding the dynamics in asymptomatics will be useful in any planned community surveillance or reactive case detection.

The focus on the clinical cases within the analysis was initially chosen so that it aligned with the population of individuals that would be sampled within the WHO protocol. However, we did also look at the dynamics within the asymptomatics when conducting the analysis for Figure 2 and found very similar patterns. This was encouraging as the data from the DRC that we used to see if our model predictions were in agreement with real data was taken from mostly asymptomatic individuals. We have included though an additional plot, Figure 1—figure supplement 1, which looks at the mean proportion of asymptomatic infections that are:

1) Only infected with *pfhrp2*-deleted parasites

2) Only infected with wild type parasites

3) Infected with both *pfhrp2-*deleted and wild type parasites

This plot also considers these proportions within clinical cases as well, which will hopefully clarify a later point about the relationship between monoclonal/multiclonal infections.

- The study model pfhrp2 deletions but no consideration is made about the effect of the type of treatment driving selection. There might be a temporal and spatial variability at this regards that has not been considered?

We do not include a selective advantage to *pfhrp2* gene deletion (apologies again for the error mentioned at the beginning of our response) and so we would not expect there to see a temporal variability in the selection pressure. If we did consider this then there would definitely be a temporal element, with the increase in the absolute number of people who seek treatment (we assume a constant proportion of people with a malarial fever seek treatment) during periods of higher transmission causing an increase in the prevalence of the *pfhrp2* gene deletion. It was because of this reason that we decided not to model selection, so that we could exclude this effect of selection and be more confident that the dynamics seen are due to the fluctuations in individuals being only infected with *pfhrp2* deleted parasites.

*- The link between transmission intensity and multiplicity of infections is clear. However, I find confusing the relationship between monoclonal/multiclonal infections and pfhrp2 deletions prevalence and selection. I think this should be elaborated further and possibly modeled?

We agree that the relationship between monoclonal/multiclonal infections is unclear, and so we hope Figure 1—figure supplement 1 described above helps clarify this.

- Previous studies indicated that PfHRP3 may play a role in the performance of PfHRP2-based RDTs. Do authors have data on pfhrp3? Apart of pfhrp2 deletions, could other sequence differences contribute to lower sensitivity of RDTs?

Attempts to quantify role that PfHRP3 has towards yielding a positive RDT result have been previously made, and we use within our modelling the assumption that a positive RDT will be produced in 25% of cases due to PfHRP3 cross-reactivity, which we sourced from Baker et al., 2005. We make reference to this study in the original paper, but not in this paper and so we have added this reference to the end of the first paragraph under the “Characterising the impact of seasonal transmission intensities upon *pfhrp2* deletion prevalence” section of the Materials and methods.

The latter point is, however, particularly interesting as we do believe that a full deletion of the *pfhrp2* gene is not essential in reducing the detection sensitivity of a PfHRP2-based RDT. The study by Baker et al. listed above looks into the impact of amino acid repeats within *pfhrp2* on the detection sensitivity of PfHRP2-based RDTs. It would definitely be of interest, however outside the scope of this study, to create an updated map of the *pfhrp2/3* genetic variants and compare it to reports of RDT performance within sub-Saharan Africa.

Reviewer #3:This paper provides novel insights into an issue of practical public health importance. The results are interesting, and deserve to be disseminated and understood. In order to achieve this fully, the paper would benefit from greater clarity in some areas. Elements of the story which are perhaps viewed as self-evident by the authors may not be self-evident to readers, and are key to interpreting the paper and its results.This paper adds seasonal variation to an individual-based model simulating prevalence of pfhpr2-del strains and false negative results in a population over time,. I have not attempted to check the original model, but the amendment shown in the current paper seems correct. Could the authors provide an updated version of the pseudo-code documentation reflecting the updates?

We have provided an updated version of the pseudocode to reflect the updates made.

WHO guidelines recommend a transition from HPR2-based RDTs to alternatives when the prevalence of false-negatives due to pfhpr2 deletion exceeds 5%, and specify survey protocols to test for this. This paper focusses on potential biases in the survey results arising from variation due to effects of seasonality and transmission intensity. Since it is central to the paper's premise, a brief explanation of the WHO survey protocol is needed, with an explicit explanation of the links between the simulation outputs and the values measured in the protocol. I think the relevant values are all present in the paper, but their meaning and relationships could be more clearly explained.

The overview of the WHO survey protocol is given within the now lengthened third paragraph of the Introduction. We have also added further clarification in the last paragraph of the Materials and methods about how our model outputs represent the population that will be sampled as part of the WHO survey protocol:

“Each repetition was subsequently simulated for 2 further years, with 7,300 individuals seeking treatment sample from each 8-week interval. […] For each sample the proportion of false-negative PfHRP2-based RDTs due to *pfhrp2/3* gene deletions was recorded.”

Can the authors clarify the basis on which the 5% threshold value was selected by WHO? The bias discussed in the paper may have different implications depending on whether the key comparator is the underlying prevalence of pfhrp2/3 gene deletions or the annual average proportion of pfhrp2/3-del false negatives. Is there any potential to add some discussion about the implications of this study for the WHO threshold value, for example whether specific values could be specified for particular seasonality and transmission-intensity contexts?

The 5% was calculated during the design of the protocol, and represents the difference in sensitivity between the HRP2-based RDTs vs. non HRP2-based RDTS. The justification is listed as follows in the WHO technical protocol:

“A threshold of 5% was selected because it is somewhere around this point that the proportion of cases missed by HRP2 RDTs due to non-hrp2 expression may be greater than the proportion of cases that would be missed by less-sensitive pLDH- based RDTs”

We agree, however, that there is a difference between the underlying prevalence of the gene deletion and the annual average proportion of false negative RDTs due to *pfhrp2/3* deletions. Ideally samples that are collected as part of the WHO protocol would also be assessed for their multiplicity of infection, and then we would also be able to work out the true deletion frequency for a given region. We could then predict at what transmission intensity that region would expect to see 5% of RDTs yielding false negatives due to *pfhrp2/3* deletions. Unfortunately this reflects a substantially larger amount of lab work given the number of samples that are likely to be collected as part of the survey as it is.

We have added some discussion about this to the end of the third paragraph of the Discussion as follows:

“It will, however, be possible after the samples have been collected to estimate the likely frequency of *pfhrp2* gene deletions by incorporating estimates of the multiplicity of infection within the sampled population. This frequency could then be used to predict the bias in future estimates, as well as estimating how the prevalence of false-negative RDT results due to *pfhrp2/3* gene deletions will change if the prevalence of malaria changes.”

The text regarding assumptions about selection and fitness (copied below) is confusing. False negative RDTs and consequent treatment choices reflected in the model will inherently exert selection, which seems to conflict with statements in the text.'Additionally, there was no assumed fitness cost or selective advantage associated with pfhrp2 gene deletion, i.e. individuals who are only infected with parasites with pfhrp2 gene deletions are assumed to yield a false-negative RDT result This decision allowed us to control for selection within our investigation. This ensures that the dynamics observed are only due to seasonal variation in transmission intensity, and not due to an increase in the frequency of pfhrp2 gene deletions due to a selective advantage by evading diagnosis. As a result, when reporting individuals who are pfhrp2-negative we assume that 25% of individuals who are only infected with pfhrp2-deleted parasites will still be pfhrp2-positive due to the cross reactivity of PfHRP3 epitopes causing a positive PfHRP2-based RDT result.'

Our apologies again for this error. This has been addressed in the opening section of our response.

[Editors' note: further revisions were requested prior to acceptance, as described below.]The manuscript has been improved but there are some remaining issues that need to be addressed before acceptance, as outlined below:As you know, there was some confusion in this paper, as the original submission indicated that model did account for selection for pfhrp2 mutants, but the subsequent correspondence indicated that the model didn't.While we understand that there is some value in considering the situation where the frequency of pfhrp2 deletions is not affected by selective forces (i.e. delayed treatment), clearly selective forces are likely to be acting in most settings and following consultation the consensus was that completely removing this real-world effect from the model was hard to justify. Therefore in addition to the analysis that has been done, the authors should add additional work where they do what they originally said they had done i.e. including a model where there is a selective advantage for pfhrp2 deletion changes/mutants as originally indicated.The authors appear to be assuming that the intended meaning of the 5% threshold is the average proportion of HRP2 RDT results for patients infected with *P. falciparum* which are false negatives caused by pfhrp2/3 gene deletions, during a given year. The Discussion and conclusion then focus on differences between the prevalence of false negatives at specific timepoints during a year vs the average prevalence value over the year. It would hugely improve the clarity of the paper to state this assumption explicitly and early in the text. It is also necessary to demonstrate using information in the WHO documentation that this is in fact the intended definition of the WHO 5% threshold. Without unequivocal evidence that this is the precise meaning of the threshold value intended by the WHO, then the use of terms such as 'bias', 'overestimate', 'underestimate' etc. is unjustified throughout.If it is not clear that the threshold is defined as an annual average, then the paper's message needs to change slightly. By indicating the extent to which the prevalence of false negatives can vary seasonally, even when the prevalence of gene deletions is constant, the results presented here indicate that a conscious choice about this aspect of the definition is very important. Should the threshold represent the acceptable maximum prevalence of false positives, or should it be the annual average. In either case, the results can inform strategies for applying the protocol in ways most likely to identify the required value.

Thank you to the reviewers and the editor for the reviews. The two major comments concerning selection and the decision to use an annual average measure are useful and we agree are important to consider. In response we have conducted the model simulations again with the assumed strength of selection used within the original study. These predictions have been included as new supplementary figures (Figure 1—figure supplement 2, Figure 2—figure supplement 1). We have also more clearly detailed why we are considering an annual average earlier in the Introduction, and have added two additional paragraphs in the Discussion exploring these comments. As the reviewers have pointed out, the annual average proportion of false-negative RDTs due to *pfhrp2/3* deletions will only be constant over time if the frequency of *pfhrp2/3* deletions are constant, i.e. under negligible selection pressure. Because of the connection of the reviewers’ two major comments (the decision to keep the frequency of *pfhrp2* deletions constant over time and the comparison of samples collected within time periods to the annual average), we wanted to address these comments firstly here in the following discussion before responding further to specific comments.

One reason for not wanting to explore selection was because this would lead to an increase in *pfhrp2* deletion, which would make it harder to observe patterns solely due to seasonal changes in transmission intensity and not due to the impact of a selective pressure. However, the true strength of any selective pressure is unknown – as raised by reviewer 3 and there are many factors that could affect how frequently individuals are only infected by *pfhrp2* deleted parasites. In addition, the technical guidance seeks to identify regions with a sufficiently high frequency of *pfhrp2/3* deletions causing false-negative RDTs (above 5% PfHRP2-negative RDTs due to *pfhrp2/3* gene deletions) as these regions will have an increase in misdiagnosed cases compared to if a non-HRP2 RDT was used. Decisions made in light of these findings are made irrespective of any assumptions about the selective pressure, except that is greater than or equal to 0, i.e. it will mean that a switch to a non-HRP2 RDT is correct even if there is no selective pressure that would increase the proportion of false-negative RDTs due to *pfhrp2-*deletions.

If we assume that there is a negligible selective pressure then the frequency of *pfhrp2* gene deletions (ignoring fade out and sufficiently large populations) will remain constant over the time period of monitoring for *pfhrp2/3* deletions. This period is defined on page 12 of the WHO technical guidance as either two years if the 95% CI for the proportion of *P. falciparum* cases with false-negative HRP2 RDT results due to *pfhrp2/3* deletions is less than 5%, or one year if it does include 5%, which is quite likely given the size of the binomial confidence interval for 370 samples. With this assumption of no selection then any measure of this proportion collected at one 8-week time period is hoped to be representative for the period of monitoring, which would thus represent the annual or biennial average (with the biennial average being the same as an annual average if there is no selection). This was why we initially were interested in looking at the annual average given our assumption of no selective pressure.

If there is a selective pressure, however, then any measure is indicative of the proportion within that time period, which would only be the same as the annual average if the strength of selection tended to zero (i.e. the no selection scenario). However, with a strong selective pressure this proportion is likely to increase. Importantly, we do not know precisely by how much and thus we do not wish to make predictions about whether a measure made in a given 8-week period is representative of that region’s average proportion in the annual or biennial period of monitoring. It is for this reason we think the recommendation for follow up studies made by the WHO in the technical guidance is very sensible. This is also because if there is a strong selective pressure but the seasonality profile of the region and the 8-week period chosen result in a measure that underestimates the annual average for the year after sample collection, then it is likely that the follow up study will correctly detect this increase. If the 8-week period chosen leads to an overestimate of the annual then the decision to switch RDT should not be considered as incorrect as it is likely that if the use of HRP2-based RDT was continued then the selective pressure would have meant a switch would have been needed at the end of the period of monitoring.

It is only in situations where there is a negligible selective pressure that you could make a decision that would actually be incorrect as opposed to premature. The regions identified in this way (Figure 4) would also likely have a negligible selective pressure as they are largely areas with both a high transmission intensity and high seasonality.

We hope the above clarifies why we were both interested in looking at areas with no selective pressure and considering annual averages. We acknowledge that this is definitely not clear nor obvious from the manuscript as it was written. To address this we have extended the Introduction to clearly lay out early what the measure we are recording is (the proportion of false-negative RDTs due to *pfhrp2/3* deletions) and why we are comparing this to the annual average, i.e. due to the period defined for follow up monitoring. We agree that this comparison is open to interpretation and so we have removed the use of terms such as bias or unbiased, and have presented the results in terms of whether they are representative of the annual average proportion. We also have added Figure 5 (referenced in the fourth paragraph of the Discussion), which is a new paragraph in which we discuss the assumption we are making about looking at the annual average as well as why we are focussing on the seasonal dynamics with the frequency of *pfhrp2* deletion fixed. This figure describes the above discussion through considering 2 scenarios: a seasonal setting with selection and without selection. In this diagram we hope to show clearly for readers why we are focussing our study on the assumption that *pfhrp2* deletion frequency is not changing over time and subsequently why we would compare measures collected in 8-week periods to the annual average.

Thank you again to the reviewers for their incredibly helpful comments and we hope our responses to the specific comments have addressed the outstanding points satisfactorily.

Because of the extensive nature of the requested revisions and clarifications which cannot easily be summarized, more extensive comments from both reviewers are appended below. All substantive points should be addressed satisfactorily as we are unable to extend the review process beyond this next revision.Reviewer #2:The manuscript has improved and authors addressed most of my comments satisfactorily. I have however a few additional comments on the revised manuscript and rebuttal letter which I feel would require additional clarification.- Exceptionally unhelpfully, the use of "false-negative" should be "positive" here. We carried out all our simulations with the assumption that individuals who are only infected with pfhrp2 gene deleted parasites will still be treated. As such, the gene deleted parasites behave exactly the same as the wild type parasites.I am afraid I don't fully follow this reasoning. My understanding is that the motivation of the study was that pfhrp2 gene deleted parasites could be indeed misdiagnosed and so simulations should treat them as false negatives (Introduction, second paragraph). If simulations threat those as positive, how could the model effectively estimate the rate of misdiagnosis and the seasonality in such estimate? May I have missed something?Besides, I don't think that the reviewers have actually addressed the real concern that came with their original consideration that false negative RTD pfhrp2 deleted parasites would allow them to control for selection.

Thank you for bringing this up, and we hope the following explanation and changes make it clearer how we simulated this. To clarify how we were conducting the earlier simulation in which no selective pressure was assumed, we assumed individuals who were infected with only *pfhrp2* deleted parasites would be correctly treated, which removed the selective pressure. However, when reporting on the rate of misdiagnosis we would regard these individuals as misdiagnosed as they are only infected with *pfhrp2* deleted parasites. Thus we are reporting the frequency of misdiagnoses that would have occurred throughout the time period for a given frequency of *pfhrp2* gene deletions, while controlling for any increases in *pfhrp2* gene deletions. The advantage of conducting simulations in this way is that we are more certain that dynamics observed in the frequency of misdiagnoses are due to seasonal fluctuations in the prevalence of individuals only infected with *pfhrp2* deleted parasites, and not due to increases in the population frequency of *pfhrp2* deletions. We have added text to clarify this in the Materials and methods section as follows:

“As a result, when reporting the proportion of clinical cases that were misdiagnosed resulting from a false-negative PfHPR2-negative RDT we are reporting the proportion of cases that are infected with only *pfhrp2-*deleted parasites, i.e. individuals who would have been *pfhrp2*-negative and subsequently misdiagnosed.”

To make this clearer for the reader, this is further expanded upon in the opening paragraph of the Results:

“We initially assumed that the frequency of *pfhrp2* deletions was not increasing over time before considering scenarios in which the selective pressure for *pfhrp2* deletions causes an increase in the population frequency of *pfhrp2* deletions. This decision allowed for the impact of seasonality on the proportion of clinical cases that are *pfhrp2*-negative to be isolated, before allowing comparisons to scenarios in which the proportion of clinical cases that are *pfhrp2*-negative is increasing also due to changes in the population frequency of *pfhrp2* deletions.”

- Related to the same issue above, and in response to my comment, authors replied:"We do not include a selective advantage to pfhrp2 gene deletion (apologies again for the error mentioned at the beginning of our response) and so we would not expect there to see a temporal variability in the selection pressure. If we did consider this then there would definitely be a temporal element, with the increase in the absolute number of people who seek treatment (we assume a constant proportion of people with a malarial fever seek treatment) during periods of higher transmission causing an increase in the prevalence of the pfhrp2 gene deletion. It was because of this reason that we decided not to model selection, so that we could exclude this effect of selection and be more confident that the dynamics seen are due to the fluctuations in individuals being only infected with pfhrp2 deleted parasites."The selective advantage comes with pfhrp2 gene deletion individuals being misdiagnosed and not getting treatment. If you consider those as positive you remove selection but this is not reflecting any more the reality of the situation.

We agree with the reviewers that a selective pressure is likely and that the presence of a selective pressure better represents the biological realism of the dynamics of *pfhrp2* gene deletions in most settings. We have conducted these additional model simulations in the revised manuscript as supplementary figures (Figure 1—figure supplement 2 and Figure 2—figure supplement 2). In these simulations, the selective pressure used in the original study was included, and the same simulation settings were explored.

The included supplementary figures for Figures 1 and 2 that consider selection show that there is a significant increase in *pfhrp2* deletions over the time period studied, in particular in the low transmission setting considered with low seasonality the frequency of *pfhrp2* gene deletions doubled from 6% to over 12% on average after two years (Figure 1—figure supplement 2Q). As a result, for this setting there is a systematic increase in the proportion of false-negative PfHRP2 RDTs within clinical cases within 8 week periods (Figure 2—figure supplement 1B). This manifests in the observation that you would expect estimates collected at the end of the calendar year (because we started our simulations in January) to overestimate the prevalence of false-negative RDTs due to *pfhrp2*-deletions when compared to the annual mean over the time period shown. However, this comparison is arguably unsuitable and perhaps should be compared to the annual prevalence after sample collection, i.e. what is the mean for the period of monitoring. We, however, do not know that true strength of selection and would feel it incorrect to suggest what this would be, whereas we are confident that the results presented when the frequency of *pfhrp2* deletions is constant are able to demonstrate which time periods are most likely to be systematically above or below the annual average.

We hope that the inclusion of these additional figures and the discussion made here have helped to clarify the above two issues. We have included a discussion of this in the third, fourth and fifth paragraph of the Discussion, which also includes some of the earlier discussion at the top of this response.

- About the relationship between monoclonal/multiclonal infections and pfhrp2 deletions prevalence and selection.I thank the authors about including a supplementary figure, but could it be possible to clarify further the relationships in the text?, saying that the relationship is unclear is not of much help.

Thank you for highlighting this as reading it back the supplementary figure wasn’t actually linked or the relationships clarified. The second Results paragraph has a longer description about why lower transmission settings have an increased chance of individuals being only infected with *pfhrp2-*negative parasites, and the supplementary figure is now linked here:

“This observation is attributable to the lower rate of superinfection in low transmission settings. The lower rate of superinfection reduces the number of polyclonal infections and increases the chance that an individual is only infected with *pfhrp2*-negative parasites (Figure 1—figure supplement 1).”

- About my comment "The regions identified were areas with both a low prevalence of malaria and a high frequency of people seeking…" Were these the only factors?"To which authors responded "These were the only factors we looked at within our modelling study".I don't find this reply satisfactory. I know they modelled only those, but my comment was more a recommendation so it is acknowledged somewhere in the Introduction or the Discussion whether they could be other factors that have not been considered and have been shown or suggested to influence the misdiagnoses.

Thank you for raising this again and apologies for misunderstanding. We have added the following sentence to the end of the second paragraph of the Introduction highlighting that there are other factors that could affect the rate of selection and the number of misdiagnoses made:

“The precise strength of selection, however, is not known with other factors such as the rate of non-malarial fevers and non-adherence to RDT outcomes likely to impact the number of misdiagnosed cases receiving treatment.“

Reviewer #3:The author's clarifications make sense and are helpful. However, my improved understanding of the authors' intentions and the results and conclusions presented in the paper has generated some additional questions and comments. I still feel that the paper would benefit from greater clarity.My understanding is that the key values being considered are;1) The proportion of HRP2 RDT results for patients infected with *P. falciparum* which are false negatives caused by pfhrp2/3 gene deletions at a given timepoint.2) The average proportion of HRP2 RDT results for patients infected with *P. falciparum* which are false negatives caused by pfhrp2/3 gene deletions, during a given year.3) The proportion of *P. falciparum* parasites in a given region which have pfhrp2/3 gene deletions.4) The 5% threshold in the WHO guidelines.It would be incredibly helpful if the authors could provide a precise definition for this, as the various wordings I have found so far in the WHO protocol and information note are open to interpretation regarding whether the 5% is intended to represent: a) The proportion of HRP2 RDT results for patients infected with *P. falciparum* which are false negatives caused by pfhrp2/3 gene deletions; orb) The proportion of *P. falciparum* parasites in a given region which have pfhrp2/3 gene deletions.Part of a full definition for this value is the assumed timing. A quick review of the WHO documentation does not immediately yield any specific information about assumed timings, an absence which would be consistent with an assumption that the rate is effectively constant through a season, or might equally mean that the relevant value is that at the time of sampling.

Thank you for the attention to the clarity regarding the model outputs that are discussed in the manuscript. We agree that we have not been consistent with which values we are referencing in the text. The main outcome that we are considering is #2 in the list above. The decision to compare to the annual average is based on the period of monitoring recommended in the WHO technical guidance on page 12, in which follow up studies should be conducted against after one year if the findings are inconclusive, i.e. the 95% CI for the proportion of false-negative RDTs due to *pfhrp2/3* deletions includes 5%. We have made this clearer in the text, with this assumption made clearer in the third paragraph of Introduction by including the specific equation listed in the WHO technical guidance as follows:

“In February 2018, the World Health Organization (WHO) issued guidance for national malaria control programmes on how to investigate suspected false-negative RDTs with an emphasis on *pfhrp2/3* gene deletions. (World Health Organization, 2018b). The primary study outcome to be calculated in the guidance is as follows:

Proportion of *P. falciparum* cases with false-negative HRP2 RDT results due to *pfhrp 2/3* deletions = # of confirmed *falciparum* patients with *pfhrp2/3* gene deletions and HRP2 RDT negative results / # of confirmed *P. falciparum* (by either RDT or microscopy)"

In the paper, the authors appear to be assuming that the intended meaning of the 5% threshold is the average proportion of HRP2 RDT results for patients infected with *P. falciparum* which are false negatives caused by pfhrp2/3 gene deletions, during a given year (item 2 in the list above). The Discussion and conclusion then focus on differences between the prevalence of false negatives at specific timepoints during a year vs the average prevalence value over the year. It would hugely improve the clarity of the paper to state this assumption explicitly and early in the text. It is also necessary to demonstrate using information in the WHO documentation that this is in fact the intended definition of the WHO 5% threshold. Without unequivocal evidence that this is the precise meaning of the threshold value intended by the WHO, then the use of terms such as 'bias', 'overestimate', 'underestimate' etc. is unjustified throughout.If it is not clear that the threshold is defined as an annual average, then the paper's message needs to change slightly. By indicating the extent to which the prevalence of false negatives can vary seasonally, even when the prevalence of gene deletions is constant, the results presented here indicate that a conscious choice about this aspect of the definition is very important. Should the threshold represent the acceptable maximum prevalence of false positives, or should it be the annual average. In either case, the results can inform strategies for applying the protocol in ways most likely to identify the required value.

Thank you for raising this issue of interpretation. We agree that the technical guidance is open to interpretation about whether the estimates collected should be reflective of that point in time or should be representative of the annual average. We have assumed the latter, which was based on the period of monitoring detailed above and on our assumption of a negligible selective pressure resulting in a constant deletion frequency. To make this clearer we have extended text in the second last paragraph of the Introduction to clarify that the comparisons to the annual average are chosen to reflect the established monitoring scheme:

“The 8-week interval permits for a rapid turnaround and allows for efficient investigations and policy responses. […] Subsequently, any recorded estimate may not be predictive of the number of cases that may be misdiagnosed due to *pfhrp2/3* deletions in the years between sampling intervals.”

However, we do acknowledge that this is open to interpretation and as such we have replaced our use of the terms biased and unbiased and refer to whether the reported estimates are representative of the annual average. We have discussed at length this in the fifth paragraph of the Discussion, in which we discuss this assumption in the context of scenarios in which the proportion of false-negative RDTs due *pfhrp2/3* deletions is increasing overtime, because these setting make the comparison to the annual average trickier. However, we conclude this paragraph by arguing that it is more prudent to be concerned with scenarios under the assumption that there is a negligible selective pressure, in which case the comparison to the annual average is justified. Our justification is demonstrated in an additional figure (Figure 5), with the following text at the end of the fourth paragraph of the Discussion raising this justification:

“However, we believe that it is more important to focus on the assumption that the strength of selection is negligible (see Figure 5). […] In areas with a selective pressure it is still possible to incorrectly estimate the annual average for the following year, however the presence of the selective pressure is likely to mean any decision made is simply premature as the frequency of *pfhrp2/3* deletions and subsequently false-negative PfHRP2 RDTs will increase over time (Figure 5B).”

There is also some confusion in the text between the prevalence of false positives results, and the prevalence of the gene deletion, with the text referring to change of RDT being triggered by an incorrect assessment of the prevalence of gene deletions (e.g. Introduction, fourth paragraph), suggesting that the authors may in fact be defining the threshold value as equal to value 3 in the list above.

Thank you for highlighting these inconsistencies, as they do make the messaging harder to follow. We agree we were not consistent in the prevalence values mentioned, and we have addressed these below.

These are key to the meaning and the implications of the work presented here, and clarity about what is being assumed or referred to is crucial to allow the text to tell its story clearly, and to make it easy to assess the consistency of that story. Confusing references to different prevalence values in the text should be reviewed and resolved wherever they arise throughout the text, including some specific instances detailed below.Detailed comments:Introduction, third and fourth paragraphs: In the third paragraph of the Introduction the authors give a definition of the WHO threshold value as being the prevalence of false negatives caused by pfhrp2/3 gene deletions. However, in the fourth paragraph of the Introduction they suggest that incorrect assessment of the prevalence of pfhrp2/3 gene deletions could drive the decision to switch to non HRP2 RDTs. Is there another mechanism in the WHO guideline in addition to the 5% false negatives threshold which would drive a change of policy based on gene deletion prevalence rather than false negative RTD prevalence?'The protocol in this guidance details how to estimate the local prevalence of false-negative PfHRP2-based RDTs due to pfhrp2/3 gene deletions and recommends that a national change to non PfHRP2-based RDTs be made if the estimated prevalence is above 5%.''the timing of the 8-week interval chosen within a transmission season could lead to bias in the sampled prevalence of pfhrp2/3 gene deletions. An overestimation of the true prevalence of pfhrp2/3 gene deletions could result in a switch to a less sensitive RDT'

Apologies for the confusion. This has been changed to refer to the proportion of *P. falciparum* cases with false-negative HRP2 RDT results due to *pfhrp2/3* deletions throughout.

Results, first paragraph and similar elsewhere in text: 'In a moderate transmission setting, a clear seasonal pattern is predicted (Figure 2C), with sampling at the beginning of the transmission seasons resulting in significant overestimation of the true proportion of false negative RDTs..''true' is not adequately defined to be used here in this way. It might legitimately be assumed to mean the population prevalence of false-negative RDTs at the time of sampling. What is meant here, I think, is that sampling at the beginning of the transmission season is expected to give a value higher than the true average value for the year.

Agree this is unclear and we have removed the use of “true” throughout and have been clearer that we are referring to the annual average, which we have detailed early on in the manuscript in the Introduction.

Introduction, last paragraph, Figure 4 description and title, Results, last paragraph.Introduction, last paragraph and figure description indicate that the values used to generate Figure 4 are the gene deletion prevalencesResults, last paragraph and implication of contents of plot indicate that the plot is based on prevalences of false negative values.Results, first and last paragraph and Discussion, first paragraph and similar elsewhere in text – 'biased' and 'unbiased' are a mathematical terms with specific meanings and it is not clear that those meanings are correctly applied here and elsewhere in the text. It would be better to replace them with other terms unless the mathematical meaning is genuinely indicated.

Thank you for highlighting this area of confusion. We have removed these terms and have consistently referred to measures as being either representative or unrepresentative of the annual average proportion of false-negative RDTs due to *pfhrp2/3* deletions.

Discussion, first and second paragraphs. These paragraphs both begin by describing the research presented in the manuscript as relating to estimates of prevalence of pfhrp2 gene deletions. The remaining text all seems to actually describe the results regarding the prevalence of false positive HRP2 RDT results, but the first sentences mean that it all reads as discussion of the gene deletion prevalence.'This research characterises the potential for surveillance in highly seasonal areas within sub-Saharan Africa to produce biased estimates of the prevalence of pfhrp2 gene deletions. These findings highlight the impact of both the seasonal timing and…''Our modelling predicted that there would be increased observation of pfhrp2 gene deletions after periods of lower transmission and within younger individuals…'

Apologies for these. We have changed these to refer consistently to the proportion of false-negative RDTs due to *pfhrp2* deletions throughout. Any mention explicitly to changes in the frequency of the gene deletions is now made in the context of how it will impact the proportion of false-negative RDTs.

Discussion, first, third and fourth paragraphs. 'However, the true prevalence of parasites with a pfhrp2 gene deletion in each administrative region is fundamentally unknown, and as such, our results should not be interpreted as predictions of the bias in future sampled estimates of pfhrp2 deletion. They should instead be used to support surveillance efforts and to reinforce the need for longitudinal measures of pfhrp2 gene deletions conducted at the same point with a transmission season.'Is this compatible with the database mentioned in the Discussion? 'To support surveillance efforts, we have published an online database detailing the optimum sampling interval as well as the sampling bias throughout the transmission season for each administrative region'

We don’t know the prevalence of deletions in all regions, however, we agree this phrasing confuses what we are trying to say here, which is simply that without knowing the fitness costs, selection pressure and the prevalence of deletions that our results are not going to be accurately predictive of how unrepresentative collected estimates may be. However, we are able to confidently indicate the time periods that are most at risk of producing estimates that do not reflect the annual average. This section has also been rewritten to better consider our assumptions about looking at the annual average in the context of our focus on simulations in which the frequency of *pfhrp2* deletions are constant over time.

'The observed prevalence of pfhrp2 deletions is higher when monoclonal infections are more prevalent, with the highest prevalence observed when sampling at the start of the rainy season as individuals are less likely to already be infected. Similarly, the observed prevalence of pfhrp2 deletions is higher in younger individuals who have lower clinical immunity, as they are more likely to present with clinical symptoms after their first infection event.'Should these two references be to prevalence of false positives rather than prevalence of pfhrp2 deletions?

Thank you again for highlighting these inconsistencies. In line with the other changes we have changed these to refer to false-negative RDTs due to *pfhrp2* deletions.

Discussion, last paragraph. This seems to be simply repeating contents of first paragraph of Discussion?

Thank you for pointing this out. We have shortened the opening paragraph of the Discussion as a result.

Subsection “Characterising the impact of seasonal transmission intensities upon pfhrp2 deletion prevalence”, last paragraph. '…fitting the frequency of pfhrp2 gene deletions in each simulation such that the true prevalence of false-negative RDT results due to pfhrp2 deletions is equal to 5%.''.. percentage of intervals that did not include the true prevalence of 5% was calculated.''true' not adequately defined, should simply say '.. the average annual prevalence..' or similar.

Thank you – we agree the use of true is unclear, and we have replaced this to now read as “annual prevalence”.

Figure 1 legend. '..In I – L and M – P the proportion of clinical cases due to pfhrp2-negative parasites is shown for both the whole population and..'Wording is confusing, does this mean cases infected only with pfhrp2-negative parasites?

Thank you for highlighting this areas of confusion. We agree and have made this clearer by changing to “proportion of clinical cases only infected with *pfhrp2*-negativeparasites”

'…the population allele frequency of pfhrp2 gene deletions, which was set equal to 6% at the beginning of each simulation..'Is the reason for or significance of the 6% value given anywhere?

Thank you for also picking up on this. The reasoning is mentioned in the Materials and methods, and relates to the 6% identified in the original study as the likely frequency of *pfhrp2* gene deletions in DRC prior to the use of RDTs. We have also added this to the opening paragraph of the Results section, which includes some description of the methodology to help improve the reader’s understanding.

'…10 simulation realisations are shown in each graph, with the mean shown with the thicker line. Lastly, the 5% threshold for switching RDT provided by the WHO is shown with the black line in plots I – P…'I think the means are shown by the black line, and the 5% by the dashed horizontal line?

Thank you for picking this up and apologies for the unclear description. This has been corrected to read:

“10 simulation realisations are shown in each graph, with the mean shown with by the black line. Lastly, the 5% threshold for switching RDT provided by the WHO is shown with the dashed horizontal line in plots I – P.”

Figure 3 legend. Should '..age and seasonality..' be '..age and transmission intensity..'?

Transmission intensity is clearer and more accurate than seasonality and has been changed. Thank you for the suggestion.

Figure 4 legend, description and title. '..pfhrp2 deletion..' should be '..false-negative pfhrp2 RDTs?..'Should also be revised as necessary to reflect assumed exact definition of threshold value.

Thank you – we have updated the caption to refer to proportion of false-negativePfHRP2RDTsdue to *pfhrp2* deletions.

pseudo codesecond line 048'// Loop through every day in simulation and calculate the seasonal curve for that day045 FOR day: = 1 TO t_max // t_max is total simulation time in days046 theta[day]:= Fourier_average +first_cosine_term * cos(2*pi*day/365) +second_cosine_term * cos(2*2*pi*day/365) +third_cosine_term * cos(3*2*pi*day/365) +first_sine_term * sin(2*pi*day/365) +second_sine_term * sin(2*2*pi*day/365) +third_sine_term * sin(3*2*pi*day/365))047 ENDFOR// Loop through every day in simulation and normalise seasonal curve for that day048 FOR day: = 1 TO t_max // t_max is total simulation time in days048 theta[day]: = theta [day] / mean(theta [1 TO 365) // normalise theta with first 365 days of theta049 IF ([day] < 0.001) // with only 1st 3 terms of Fourier used we need to check for <0050 [day]: = 0.001051 ENDIF052 ENDFORI'm assuming this is just a problem with the pseudo code, not the actual code, but that should be checked and confirmed. It seems that in the normalisation loop, the sum of theta values by which θ(n) is divided will use the normalised rather than original values for all θ(<n).Could the authors please review the pseudo code for consistency with the actual code?

Apologies for the short hand here in the pseudocode as we agree it was misleading. The normalisation is using the mean of the original values. This has been corrected through specifying the calculation of the man prior to the normalisation loop. The line numbers have all been updated as well now to reflect this extra line.